# A chimeric Mla-Pqi lipid transport system is required for *Brucella abortus* survival in macrophages

Adélie Lannoy [ID][1], Alexi Ronneau[1], Miguel Fernández-García [ID][2,3], Marc Dieu [ID][4,5], Patricia Renard [ID][4,5], Antonia García Fernández[2], Raquel Condez-Alvarez[6] & Xavier De Bolle [ID][1,7 ✉]

## Abstract

**The cell envelope of gram-negative bacteria is composed of an inner and an outer membrane. In *Escherichia coli*, several pathways mediate phospholipid transport between the two membranes, including the Mla (i.e., maintenance of lipid asymmetry) and Pqi (i.e., paraquat inducible) systems. Here, we identify and characterise in the intracellular pathogen *Brucella abortus* a complex named Mpc, which exhibits homology to both Mla and Pqi components. Mpc is required for bacterial growth under envelope stress conditions, and for survival within macrophages during the early stages of infection. Analyses of protein-protein interactions and structural predictions suggest that the Mpc complex bridges the two membranes of the bacterial cell envelope. Absence of this system results in altered lipid composition of the outer membrane vesicles, indicating that Mpc plays a role in lipid transport between the membranes. Our sequence comparisons reveal that Mpc is conserved across numerous species of Hyphomicrobiales. The discovery of this novel lipid-trafficking system expands our understanding of the diversity and evolution of lipid-transport mechanisms in diderm bacteria.**

**Keywords** Lipid Transport; Outer Membrane Biogenesis; Pqi; Mla; *Brucella*
**Subject Categories** Membranes & Trafficking; Microbiology, Virology & Host Pathogen Interaction

## Introduction

The integrity of the cell envelope is crucial for the survival and growth of bacteria. Gram-negative bacteria are characterised by the presence of an inner membrane (IM), a periplasmic space containing a thin layer of peptidoglycan (PG) and an outer membrane (OM). The IM is a bilayer of phospholipids (PLs), while the OM is an asymmetric bilayer. The OM inner leaflet is predominantly composed of PLs, while the outer leaflet lipid is mainly composed of lipopolysaccharide (LPS).

The OM functions as an essential protective and permeability barrier (Silhavy et al, 2010).

All envelope components, whether proteic or lipidic, are biosynthesised in the cytoplasm or in the inner leaflet of the IM. They are then translocated from the cytoplasm to the outer leaflet of the IM, or flipped across the IM (Silhavy et al, 2010). In *Escherichia coli*, the transport systems across the periplasm are relatively well-understood with regard to the delivery of outer membrane proteins (OMPs) (Konovalova et al, 2017; Tomasek and Kahne, 2021), lipoproteins (Grabowicz, 2018) and LPS (Giacometti et al, 2022; Okuda et al, 2016) to the OM. However, the understanding of the systems which facilitate PLs transport are still unclear (Kumar and Ruiz, 2023).

Two distinct mechanisms are involved in the transport of PLs across the periplasm. Either proteins form a bridge spanning the entire periplasm, or a shuttle in the form of a soluble carrier protein is involved. The periplasmic-spanning AsmA-like protein family, which includes TamB, YhdP and YdbH, exhibits a specific conformation in a Taco-like shape. It is characterised by a hydrophobic groove, which facilitates the transport of lipids (Cooper et al, 2025; Douglass et al, 2022; Grasekamp et al, 2023; Kumar and Ruiz, 2023; Ruiz et al, 2021; Sposato et al, 2024). Additionally, protein complexes spanning the periplasm with a channel shape have been described for Pqi (paraquat inducible) and Let (lipophilic envelope-spanning tunnel; formerly Yeb) (Ekiert et al, 2017; Isom et al, 2020). The shuttle-based system is the Mla (maintenance of OM lipid asymmetry) pathway (Malinverni and Silhavy, 2009). A common feature of the Mla, Pqi and Let systems is the presence of a protein comprising one or more MCE domains (mediator of mammalian cell entry), arranged as homo-hexamers in the structures that have been solved thus far (Ekiert et al, 2017; Isom et al, 2020; Isom et al, 2017; Nakayama and Zhang-Akiyama, 2017). The MCE superfamily of proteins is ubiquitous among diderm bacteria (Arruda et al, 1993; Chen et al, 2023a; Cooper et al, 2024; Ekiert et al, 2017; Grasekamp et al, 2023; Isom et al, 2020; Isom et al, 2017; Malinverni and Silhavy, 2009) and eukaryotic bacteria-derived organelles (Awai et al, 2006; Isom et al, 2017; Lu and Benning, 2009).

The Mla pathway is composed of the OmpC-MlaA complex in the OM, a MlaC periplasmic shuttle, that facilitates the transport of

[1]URBM and Department of Biology, Namur Research Institute for Life Sciences (NARILIS), Namur, Belgium. [2]Centro de Metabolómica y Bioanálisis (CEMBIO), Facultad de Farmacia, Universidad San Pablo-CEU, CEU Universities, Urbanización Montepríncipe, 28660 Boadilla del Monte, Spain. [3]Departamento de Ciencias Médicas Básicas, Facultad de Medicina, Universidad San Pablo-CEU, CEU Universities, Madrid, Spain. [4]URBC, Department of Biology, Namur Research Institute for Life Sciences (NARILIS), Namur, Belgium. [5]MaSUN, Mass Spectrometry Facility, University of Namur, Namur, Belgium. [6]Department of Microbiology and Parasitology, Instituto de Investigación Sanitaria de Navarra (IdiSNA), University of Navarra, 31008 Pamplona, Spain. [7]WEL Research Institute, avenue Pasteur, 6, 1300 Wavre, Belgique. ✉E-mail: xavier.debolle@unamur.be

PLs through the periplasmic space, and MlaEFDB, a highly conserved ATP-binding cassette (ABC) transporter complex in the IM. MlaD comprises a transmembrane helix and a periplasmic MCE domain and forms a ring-shaped homo-hexameric structure (Coudray et al, 2020; Ekiert et al, 2017). This system is responsible for the removal of mislocalised PLs from the OM and their subsequent transport back to the IM, which allows for the maintenance of the lipid asymmetry of the OM (Chong et al, 2015; Malinverni and Silhavy, 2009; Shrivastava and Chng, 2019). Recent studies have expanded the research of the Mla system to diverse bacterial species, including *Burkholderia cepacian* (Bernier et al, 2018), *Acinetobacter baumannii* (Powers and Trent, 2018), *Pseudomonas aeruginosa* (Zhou et al, 2021), *Bordetella pertussis* (de Jonge et al, 2022), *Stenotrophomonas maltophilia* (Coves et al, 2024) and even in a diderm Firmicute, *Veillonella parvula*, which present a Mla system lacking MlaA and MlaC (Grasekamp et al, 2023). The Pqi complex is composed of an integral membrane protein of the IM (PqiA), devoid of an ATPase domain, and a PqiB homo-hexameric oligomer. This oligomer forms a stack of three rings composed of MCE domains which is followed by a syringe-like structure of coiled-coil alpha-helices (Ekiert et al, 2017). This structure spans the periplasmic space to interact with the lipoprotein PqiC in the OM. PqiC is arranged as a homo-octameric ring, which may serve to anchor the C-terminal part of PqiB to the OM, thereby stabilising the flexibility of the coiled-coil channel of the PqiB multimer (Cooper et al, 2024). The Let complex is composed of an integral membrane protein of the IM (LetA) lacking an ATPase domain, and LetB, a homo-hexameric oligomer containing a stack of seven MCE domains. The MCE domains form a channel that is capable of mediating direct lipid transport between the IM and the OM by forming a bridge across the periplasm (Ekiert et al, 2017; Isom et al, 2020).

It is noteworthy that MCE domains were initially identified in *Mycobacterium tuberculosis* (Arruda et al, 1993) and that the structure of the MCE complex proteins, in *M. smegmatis*, was recently resolved. This Mce1 system forms an elongated needle structure that spans the cell envelope (Chen et al, 2023a). In addition, co-evolutionary analysis of domains in TGD2, a lipid transporter across cell membranes, from several plants revealed that TGD2 has an MCE domain and a helical region in the C-terminal portion that forms a needle-like shape (Yang et al, 2017). MCE domains were proposed to play a significant role in the transport of lipids and also in the virulence of bacterial pathogens (Awai et al, 2006; Ekiert et al, 2017; Malinverni and Silhavy, 2009; Nakayama and Zhang-Akiyama, 2017). Furthermore, the integrity of the membranes is also an important factor in the virulence of bacterial pathogens. The envelope structure and biogenesis of *Brucella abortus* were recently reviewed (Alakavuklar et al, 2023). *B. abortus* is a Gram-negative intracellular pathogen responsible for bovine brucellosis, a worldwide zoonosis with significant social and economic impacts (Laine et al, 2023; Moreno and Moriyón, 2006). In contrast to *E. coli*, the members of the Hyphomicrobiales order bacteria, such as *B. abortus*, *Agrobacterium tumefaciens* and *Sinorhizobium meliloti*, display unipolar growth (Brown et al, 2012). The incorporation of LPS, PG and OMPs occurs at the new pole and the division site in *B. abortus* (Servais et al, 2023; Vassen et al, 2019). Moreover, the OM is linked to the PG by covalent cross-links between the N-terminus of OMPs and the peptide stems of PG (Godessart et al, 2021). Nevertheless, the biogenesis and

turnover of the envelope in *B. abortus* remain under-investigated, particularly regarding the transport of PLs. As an intracellular pathogen, *B. abortus* encounters numerous stresses during its infectious cycle, such as antimicrobial peptides, which could potentially affect the integrity of its envelope.

In this study, we characterise the Mpc (Mla-Pqi chimeric) system comprising the only MCE domain-containing protein predicted from the *B. abortus* genome. The Mpc system is required for *B. abortus* growth in the presence of envelope stress and for survival during the first stage of macrophage infection. Structure prediction and pull-down analysis strongly suggest that the Mpc system forms a stable complex that spans the entire periplasmic space. The absence of the Mpc system generates an abnormal lipid composition of the OM vesicles, supporting its role in the trafficking of lipids between IM and OM in *B. abortus*.

# Results

## Identification of the *mpc* genes

In order to identify a system required to resist envelope stress, a transposon sequencing (Tn-seq) analysis was performed. A *Brucella abortus* 544 library ($7.8 \times 10^5$ random mutants), constructed with the mini-Tn5 (Sternon et al, 2018), was grown on rich medium with 0.015%. of sodium deoxycholate (DOC) and compared to a control condition without DOC. DOC, a detergent that affects the integrity of bacterial membranes (Urdaneta and Casadesus, 2017), was chosen over SDS and EDTA, which are commonly used in *E. coli*, because *Brucella spp.* are resistant to EDTA (Moriyon and Berman, 1982), but sensitive to DOC (Caro-Hernandez et al, 2007). To identify ORFs required for growth in the presence of DOC, the ΔTnIF variable was computed (Figs. 1 and EV1). The more negative the ΔTnIF, the more the ORF is required for growth in the presence of DOC. The genes encoding for the BepCDE RND efflux pump, which is known to be involved in DOC resistance (Martin et al, 2009; Posadas et al, 2007), were selected here (Fig. 1), thereby confirming the efficiency of the Tn-seq approach for identifying genes involved in DOC sensitivity. The genes encoding the CenKR two-component system (TCS), were identified with a ΔTnIF less than four times the genome standard deviation. This TCS involved in cell envelope integrity has been previously described in several Hyphomicrobiales (Lakey et al, 2022; Skerker et al, 2005) and *Brucella* spp (Chen et al, 2023b; Liu et al, 2012; Zhang et al, 2009). Of particular interest are the four more strongly attenuated genes (Fig. 1), with a ΔTnIF less than four times the genome standard deviation, which are found in the *mpcEFDA* operon. This operon encodes a potential system containing an MCE protein, which is involved in lipid transport (Nakayama and Zhang-Akiyama, 2017). This system has not yet been characterised in *Brucella* spp.

To confirm the phenotype suggested by Tn-seq, markerless deletion strains for *mpcE*, *mpcF*, *mpcD*, *mpcA* and *mpc* operon (collectively named *mpc* mutants) were generated in *B. abortus*. Each strain exhibited a significant growth defect on DOC at a concentration of only 0.005% (w/v), and the complemented mutant strains restored the wild-type phenotype (Fig. 2). In addition, the sensitivity to vancomycin was investigated to ascertain whether the outer membrane was destabilised. The wild-type (WT) strain and all the *mpc* mutants grew similarly in the presence of vancomycin at

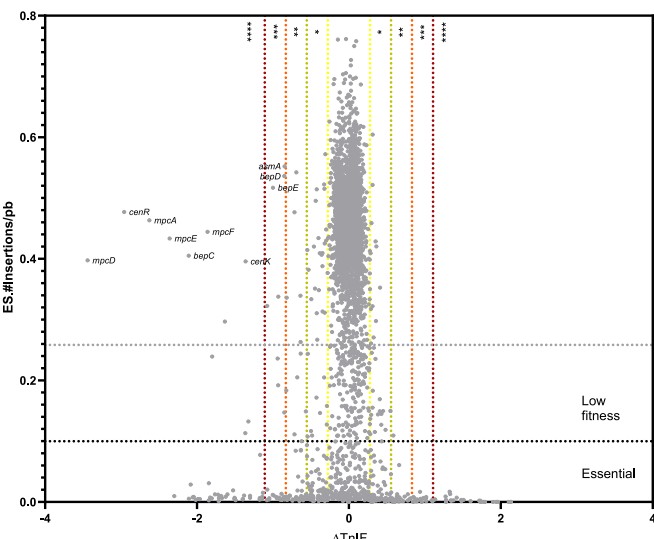

**Figure 1.  DOC-sensitive mutants according to Tn-seq.**

For each mutated gene, a Transposon insertion frequency (TnIF) is computed. Transposon insertion sites were identified through Illumina sequencing. In the control condition, there was an average of one unique insertion site every 2.62 bp, and in the envelope stress condition, every 2.48 bp, saturating the *B. abortus* genome in both conditions. Following the mapping of reads, the ES.#insertion/pb and an ES.Reads values were computed for each open reading frame (ORF), corresponding to the insertion number per bp for 80% of each ORF, by excluding the first and last 10% of the predicted coding sequence. The ES.#insertion/pb facilitates the determination of the essentiality of each ORF, with an ORF classified as essential if the value is less than 0.1, and as low fitness if the value is less than 0.285 (the mean of ES.#insertion/pb value minus 0.1). To enable a quantitative analysis of each ORF under different conditions, we calculated a transposon insertion frequency (TnIF) parameter. This frequency corresponds to the logarithm in base 10 of the mapped read numbers for the central 80% of each ORF ($\log_{10}$ (ES.Reads+1)). To identify genes required for growth in the presence of DOC (0.015%), we compared the TnIF of the stress condition to the control condition to obtain a $\Delta$TnIF for each ORF ($\Delta$TnIF = $TnIF_{DOC}$ - $TnIF_{Ctrl}$). The standard deviation (SD) of $\Delta$TnIF values for all the ORFs corresponds to 0.323. We considered ORFs with less than one SD (*;±0.323) as less required, two SD (**;±0.647) as required, three SD (***;±0.97) as highly required and four SD (****;±1.293) as required for growth on DOC. Source data are available online for this figure.

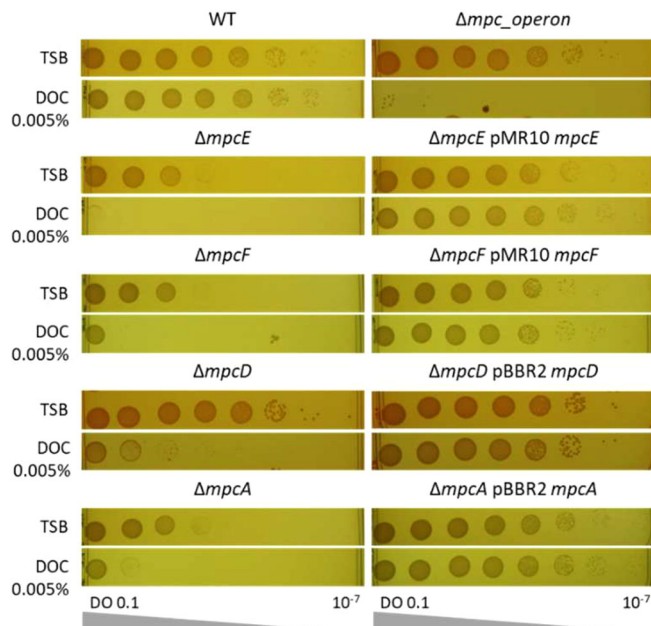

**Figure 2.  The *mpc* mutants exhibit a significant growth defect on DOC.**

A plating assay shows that *mpc* mutants are sensitive to DOC. The overnight cultures were normalised to 0.1 $OD_{600}$ and serially diluted before being plated onto TSA with or without DOC. All *mpc* mutants exhibit a growth defect on DOC at 0.005%. The complemented strains exhibit a wild-type growth phenotype. All mutants except $\Delta$*mpcD* also display a growth defect on TSA without DOC. Source data are available online for this figure.

1000 μg/mL, compared to the rich medium. This finding suggests that the *mpc* mutants are unable to adapt to envelope stress, at least in the presence of DOC.

## The Mpc system is required for survival in macrophages

To test if the envelope stress sensitivity of the *mpc* mutants correlates with a defect in virulence, J774.A1 macrophages were infected with these strains. The number of viable intracellular bacteria was evaluated with a colony-forming unit (CFU) count after gentamycin treatment, which kills extracellular bacteria. The WT strain has been observed to exhibit two distinct phases of infection, characterised by an initial absence of growth between 2 and 5 h post-infection, followed by a period of substantial growth, as previously reported (Deghelt et al, 2014; Starr et al, 2008). The CFU counts of the mutants were significantly lower at 2 and 5 h post-infection (Fig. 3A). At 24 and 48 h post-infection, the mutant strains remained at a lower level than the WT, but their

intracellular growth was not affected (Fig. 3A; Appendix Fig. S1). To determine the ability of the bacteria to adhere and enter host cells, a double labelling approach was employed. Firstly, the bacteria were labelled with an antibody against the surface of *B. abortus*, followed by permeabilisation of the host cells. Subsequently, a second labelling of bacteria was performed, utilising another antibody against the surface of *B. abortus*, labelling bacteria intra- and extracellular. This technique enabled the identification of the bacteria outside, inside and on the host cell. The results showed that the ability of the *mpc* mutants to adhere and enter the macrophages was very similar to the WT strain (Fig. 3B). These results suggest that *mpc* mutants present survival defects in macrophages during the first hours of infection.

## The outer membrane vesicles' lipid composition depends on Mpc

In *E. coli*, the Mla pathway is implicated in the retrograde transport of phospholipids between the outer and inner membranes (Malinverni and Silhavy, 2009; Tang et al, 2021; Wotherspoon et al, 2024). To investigate the possible function of the Mpc pathway as a PLs transporter, outer membrane vesicles (OMVs) were isolated to characterise their PLs composition. Total lipid extraction was performed on OMVs and whole cell samples from *mpc* mutants and the WT. The thin layer chromatography (TLC) on lipid extractions showed that all *mpc* mutants have less amount of cardiolipin (CL) in OMVs compared to the WT (Fig. 4A; Appendix Fig. S2). Furthermore, there appeared to be no variations

**A.**

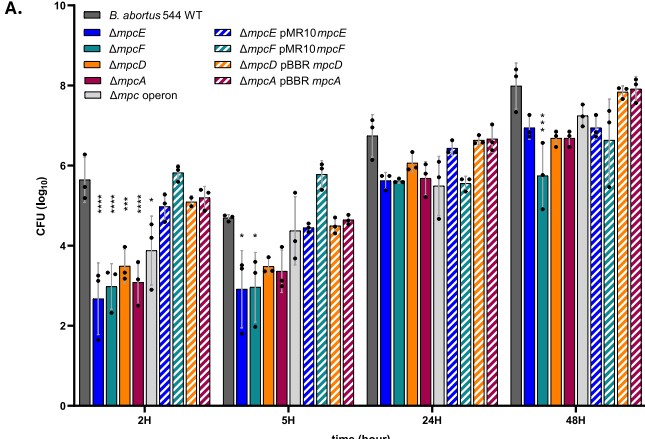

**B.**

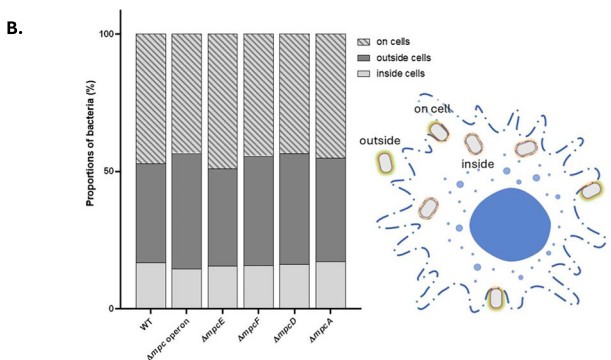

**Figure 3. The *mpc* mutants are killed during early trafficking.**

(A) Intracellular replication of WT, *mpc* mutants and the complemented strains were assessed by counting colony-forming units (CFU) at 2, 5, 24 and 48 h post-infection of J774.A1 macrophages. The data represent the mean ± SD and were compiled from three biological replicates, represented by the dots. Statistical significance between the results for a given strain and those for the WT was determined using a two-way ANOVA (not significant: $p ≥ 0.05$; *$0.05 < p < 0.01$, ***$0.001 < p < 0.0001$, ****$p < 0.0001$). The significant *P* values are $P = 1,07e-7$ for *mpcE*, $P = 2,97e-6$ for *mpcF*, $P = 5,48e-4$ for *mpcD*, $P = 8,92e-6$ for *mpcA*, $P = 1,73e-2$ for *mpc* operon at 2 h PI and $P = 1,55e-2$ for *mpcE*, $P = 2,33e-2$ for *mpcF* at 5 h PI and $P = 2,31e-4$ for *mpcF* at 48 h PI. (B) To identify the localisation of bacteria inside, outside or on the macrophages, a double labelling technique was performed. After 2 h post-infection, the cells were fixed with 4% PFA and the bacteria were labelled with a rabbit serum against *Brucella* (Deghelt et al, 2014). After the first set of labelling, the cell membrane was permeabilised with Triton X-100 0.1% and the bacteria were labelled a second time with a mouse monoclonal antibody against *Brucella abortus* lipopolysaccharide. Bacteria that were labelled only with the mouse monoclonal antibody are inside the cell, while those that were labelled with both antibodies are outside the cell. Bacteria adhering to the surface of the macrophages were found labelled with the mouse monoclonal antibody on a part of the bacterial cell only. For each strain, the number of bacteria was counted by two individuals in 108 macrophages. The mean number of bacteria is presented as a percentage, depending on their location. The proportions of each localisation type are unchanged for the mutant strains compared to the wildtype, indicating that adherence and entry of the bacteria into the macrophages was not affected, while survival of the mutant was strongly impaired. Source data are available online for this figure.

in PLs composition between the mutants and the WT in whole bacterial lysates. These findings suggested that the Mpc system may act as a PLs transport system.

To obtain more information on the OMVs' lipid composition, we analysed the total lipid composition using liquid chromatography

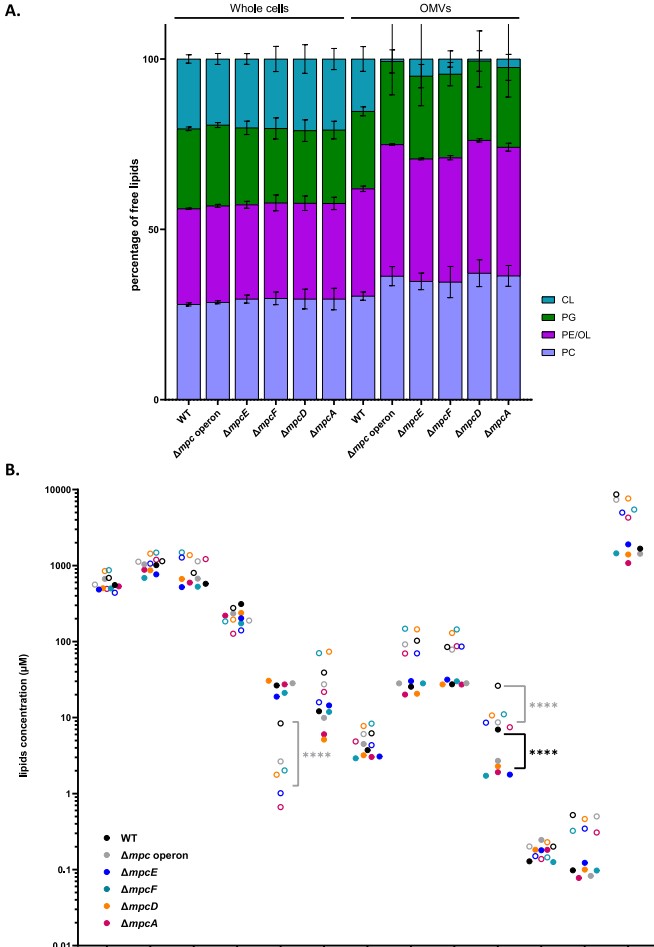

**Figure 4. Lipid composition of whole cells compared to outer membrane vesicles.**

(A) Quantification of bands of thin layer chromatography was performed using Quantity One software. The mean ± SD band intensity from three technical replicates was normalised. The stacked bars represent the percentages for each strain in both whole cells and outer membrane vesicles. PC phosphatidylcholine, PE phosphatidylethanolamine, OL ornithine lipid, PG phosphatidylglycerol, CL cardiolipin. (B) Dot plot of the phospholipid concentration (μM) identified by mass spectrometry-based lipidomic analysis on *mpc* mutants and WT strains. The whole cell samples are represented by full dots and the OMVs by empty dots. Statistical significance between the results for the five mutant strains relative to the WT was determined using a conformance test (unpaired two-tailed *t*-test) for each phospholipid concentrations **$p < 0.01$ and ***$p < 0.001$ (in grey for the OMVs and in black for the whole cells samples). PG phosphatidylglycerol, PE phosphatidylethanolamine, PC phosphatidylcholine, OL ornithine lipid, CL cardiolipin, LPG lyso-phosphatidylglycerol, acyl-PG aminoa-cyl-phosphatidylglycerol, LPE lyso-phosphatidylethanolamine, LPC lyso-phos-phatidylcholine, LOL lyso-ornithine lipid, FAHFA fatty acid esters, PA Phosphatidate of hydroxy fatty acid, FA fatty acid. The significant *P* values are $P = 4,49e-5$ for CL in OMVs, $P = 1,52e-5$ for LOL in OMVs and $P = 1,20e-5$ for LOL in whole cells. Source data are available online for this figure.

coupled to quadrupole-time-of-flight mass spectrometry, LC-MS (QTOF). The analysis revealed that the lipid composition of OMVs and whole cells are different but shares commonalities in the changes associated with the *mpc* mutants. Some differences were observed between the WT and *mpc* mutants in both whole cells and OMVs lipid extracts. The WT and *mpc* mutants differed in concentration of CL in

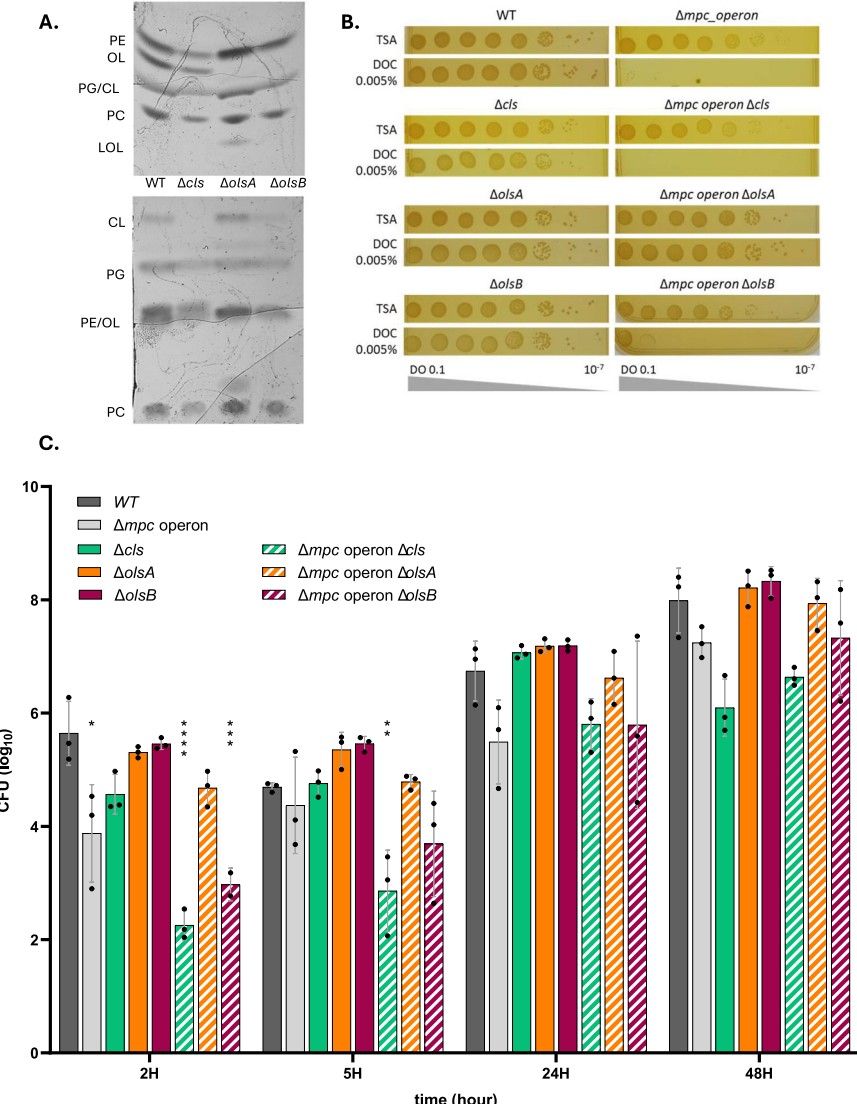

**Figure 5. The sensitivity phenotype of *cls*, *olsA* and *olsB* deletion mutants.**

(A) Thin layer chromatography (TLC) was performed on lipid extracts obtained from whole cells of lipid synthase mutants (1, WT; 2, Δ*cls*; 3, Δ*olsA*; 4, Δ*olsB*). On the top panel, the lipids were separated by using a solution of TCM, MeOH and $H_2O_{dd}$ (14:6:1) and in the bottom panel of TCM, MeOH and AcOH (13:5:2). PC phosphatidylcholine, PE phosphatidylethanolamine, OL ornithine lipids, PG phosphatidylglycerol and CL cardiolipin. (B) The plating assay on *cls*, *olsA*, *olsB* and *asmA* mutants, *mpc* operon mutant and double mutants on DOC. All the mutants with deletion of the *mpc* operon present the same sensitivity to DOC at 0.005%, except for the double mutant with *olsA*, which grows like the WT strain. The overnight cultures were normalised to 0.1 $OD_{600}$ and serially diluted before being plated onto TSA with or without DOC. (C) Intracellular replication of WT, *cls* mutant, *mpc* operon mutant and the double mutants were assessed by CFU at 2, 5, 24 and 48 h post-infection of J774.A1 macrophages. The data represent the mean ± SD and were compiled from three biological replicates, represented by the dots. Statistical significance between the results for a given strain and those for the WT was determined using a two-way ANOVA (not significant: $p \geq 0.05$; *$0.05 < p < 0.01$, **$0.01 < p < 0.001$, ***$0.001 < p < 0.0001$, ****$p < 0.0001$). The significant P values are $P = 4,56e\text{-}2$ for *mpc* operon, $P = 6,64e\text{-}8$ for *mpc* operon *cls* and $P = 4,41e\text{-}4$ for *mpc* operon *olsB* at 2 h and $P\ 2,99e\text{-}2$ for *mpc* operon *cls* at 5 h. Source data are available online for this figure.

OMVs lipid extracts and lyso-ornithine lipids (LOL) in whole cells and OMVs (Fig. 4B). The lipid extracts from all mutants' OMVs exhibit a profile that differed from that of the WT, while the profiles of whole cell samples were more similar (Appendix Fig. S3). These data indicate that OMVs exhibit a notable decrease in CL in the *mpc* mutants and reveal that LOL are also less abundant in whole cells and OMVs of these mutants.

To ascertain whether the observed stress sensitivity phenotype is a consequence of the absence of CL or LOL within the OM or their

accumulation in the IM, the phenotype of a cardiolipin synthase (Cls) and ornithine lipid synthases (OlsA and OlsB) deletion mutants were subjected to analysis (Fig. 5A). The conversion of ornithine to OL occurs in two stages. The initial step is catalysed by OlsB, which adds a fatty acyl group to ornithine to form LOL. The second step is the addition of a fatty acyl group to the LOL, transforming LOL to OL through the action of OlsA (Gao et al, 2004). The deletion mutants, which no longer possess CL, LOL or OL, did not exhibit any sensitivity in the presence of DOC (Fig. 5B)

or in the infection model (Fig. 5C). This suggests that the absence of CL, LOL or OL is not sufficient to cause DOC sensitivity and attenuation in this cellular infection model. However, when combined with the *mpc* operon deletion, no increase in DOC sensitivity and no significant attenuation in J774.A1 macrophages was observed in comparison to the Δ*mpc* operon mutant, only for the *cls* and *olsB* deletions. The deletion of *olsA* in the background of the *mpc* operon deletion restores the WT phenotype in both conditions (Fig. 5B,C). This result suggests that the putative accumulation of LOL, induced by the absence of OlsA, could compensate for the absence of the Mpc system.

## Mpc could bridge the inner to the outer membrane

The *mpc* operon in the *B. abortus* genome consists of only four genes, compared to six genes encoding for the Mla pathway in *E. coli*. Structural modelling was performed to determine how this simplified system could transport PLs between both membranes (Fig. 6A). The ABC transporter present in the IM is well-conserved, with the ABC ATPase MpcF homologous to MlaF, and the integral IM ABC permease MpcE homologous to MlaE, but possessing an additional N-terminal STAS (sulfate transporter and anti-sigma factor antagonist) domain, which is homologous to the MlaB protein (Appendix Fig. S4). The 100 first amino acids of MpcD aligns with MlaD. However, the MpcD (331 aa) is almost twice as long as MlaD (183 aa). The extended C-terminal domain of MpcD is a conserved feature among several Hyphomicrobiales, with the domain being even more extended in *Agrobacterium tumefaciens*

and *Sinorhizobium meliloti*. The secondary structure prediction also exhibits conservation among Hyphomicrobiales, displaying a series of alpha-helices in the C-terminal part of the protein (Fig. EV2). This extended C-terminal has already been characterised in *M. tuberculosis* and *Veillonella parvula*, which form an elongated needle structure that spans the cell envelope (Chen et al, 2023a; Grasekamp et al, 2023). It should be noted that the IM ABC transporter is also homologous to the YrbE1A-B and TGD2-3 complexes from *M. tuberculosis* and *Arabidopsis thaliana*, respectively. The needle structure of MpcD is similar to the heterohexamer Mce1 from *M. tuberculosis* and the homo-hexamer TGD2 from *A. thaliana* (Appendix Fig. S5)(Chen et al, 2023a; Roston et al, 2012). Interestingly, the lipoprotein MpcA (encoded by the last gene of the *mpc* operon) shows homology with the PqiC protein, but not with MlaA (Appendix Fig. S4). Additionally, there is no detectable homologue for MlaC in the *B. abortus* genome. The system was named the Mla-Pqi chimeric (Mpc) system as it encodes for a hybrid system between the Mla pathway and Pqi system (Fig. 7). The AlphaFold3 multimer predictions of biomolecular interactions (Abramson et al, 2024) of Mpc proteins and some oleic acids as model lipids were conducted. As the number of monomers in the $MlaE_2F_2D_6B_2$ and $PqiA_2B_6C_8$ complexes is known (Cooper et al, 2024; Ekiert et al, 2017; Malinverni and Silhavy, 2009; Nakayama and Zhang-Akiyama, 2017), we used the same multimeric state for the Mpc complex prediction, $MpcE_2F_2D_6A_8$. The prediction for the entire complex was unsuccessful due to the coiled-coil alpha-helices of the $MpcD_6$ channel being folded over themselves (Appendix Fig. S6).

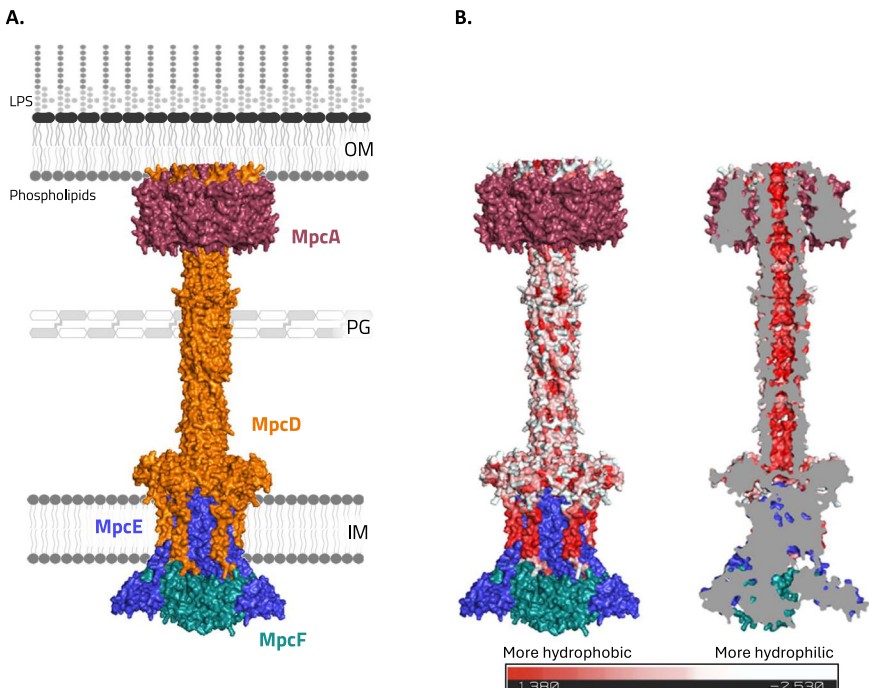

**Figure 6. The multimer predictions of the Mpc complex.**

(A) Model of the interaction between Mpc proteins predicted with AlphaFold3 multimer predictions of biomolecular interactions (Abramson et al, 2024). The structure prediction of $MpcE_2F_2D_6$ and $MpcD_6A_8$ were fused by alignment of the C-terminal part ($MpcD_{D241-N310}$) of alpha-helices of MpcD on PyMol. (B) The hydrophobicity of the full-length hexameric MpcD model was calculated using the PyMol color_h command. The MpcD model has a length of around ~250 Å with a transmembrane domain in the N-terminal part. The tunnel of this model displays a hydrophobic interior, with a diameter of around 18 Å. In the cut-away view (right panel).

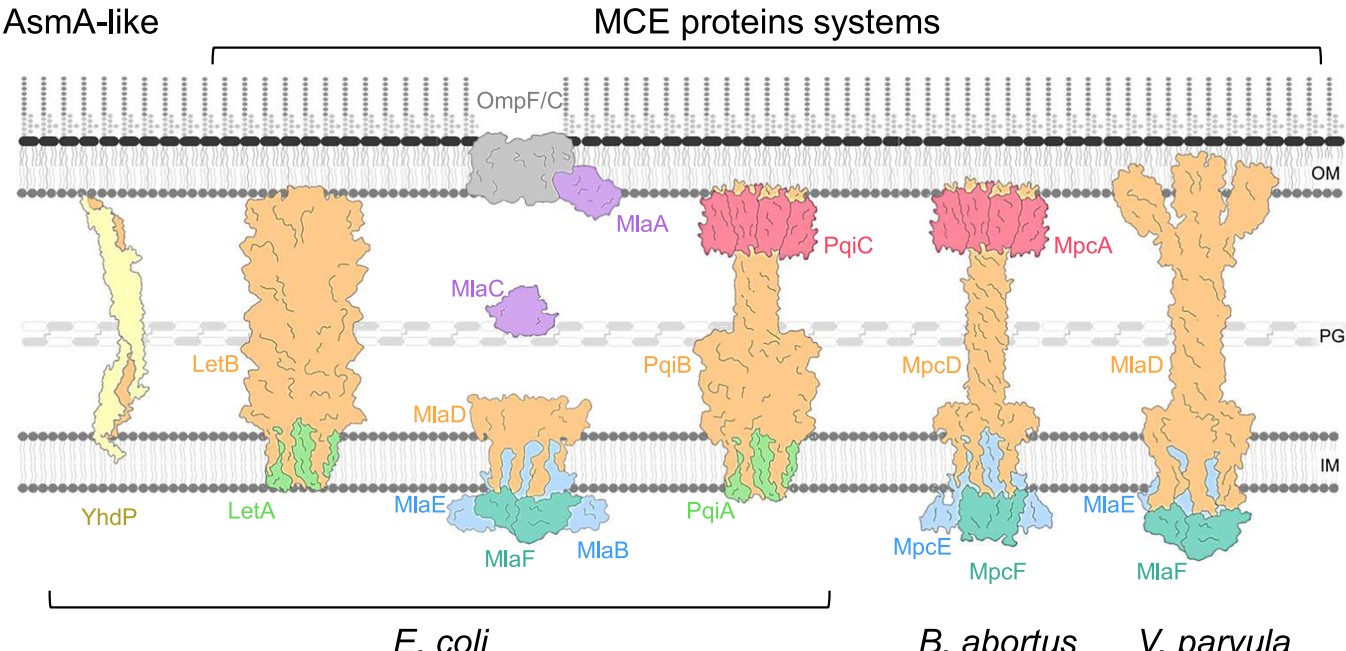

**Figure 7. AsmA-like proteins and MCE proteins phospholipid transporters.**

Schematic representations of phospholipid transporters in *Escherichia coli*. AsmA-like proteins are represented by the YhdP structure, from AlphaFold (Cooper et al, 2025). The let system, Mla pathway and Pqi system comprise MCE proteins (orange). LetB (6V0C) structure was determined by Cryo-EM (Isom et al, 2020), LetA structure is not characterised. MlaEFDB (6XBD) structure was determined by Cryo-EM (Coudray et al, 2020), MlaC (5UWA) structure was determined by Cryo-EM (Ekiert et al, 2017) and OmpF-MlaA (5NUO) structure was determined by X-ray diffraction (Abellon-Ruiz et al, 2017). Multi-MCE domain structure of PqiB (5UVN) was determined by Cryo-EM (Ekiert et al, 2017). PqiC (8Q2C) X-ray diffraction structure was obtained, and the structure of PqiABC was predicted by AlphaFold-Multimer (Cooper et al, 2024). The structure of the MpcEFDA system in *Brucella abortus*, was predicted by AlphaFold-Multimer in this study. For the Mla system homologues in *Veillonella parvula*, the structure of MlaD was predicted by AlphaFold2 and the homologues of MlaE and MlaF were identified as conserved (Grasekamp et al, 2023).

Subsequently, we proceeded to predict the interactions of $MpcD_6$ with $MpcA_8$ and, in another instance, with $MpcE_2F_2$ in the presence or not of 50 oleic acids (OLAs) that could potentially mimic interactions with lipids (Fig. EV3). The $MpcE_2F_2D_6$-50 OLAs AlphaFold3-predicted structure exhibited higher confidence for the MCE and transmembrane domains, suggesting that the IM domains may be stabilised by the presence of hydrophobic molecules (Fig. EV3). For all prediction complexes, the repeat coiled-coil alpha-helices domains exhibited low prediction confidence, which is likely due to the repeating domain (Appendix Fig. S6). The $MpcD_6A_8$ AlphaFold3-predicted structure exhibits a higher confidence in the octameric form of MpcA, but a lower confidence in the hexameric form of MpcD compared to the $MpcE_2F_2D_6$ structure prediction (Fig. EV3 and Appendix Fig. S6). A fusion of the two AlphaFold3-predicted structures is illustrated in Fig. 5A. The complex length, ~310 Å, is sufficient to span the entire periplasmic space, which has a mean size of ~30 nm (Godessart et al, 2021). Additionally, the predicted hexameric structure of MpcD reveals an internal hydrophobic channel (Fig. 6B) that may facilitate the transport of hydrophobic compounds, such as lipids.

To confirm the interaction of Mpc proteins, a pull-down assay was conducted on MpcA fused with a Strep tag, followed by analysis through liquid chromatography coupled with tandem mass spectrometry (LC-MS/MS). The strain producing MpcA-Strep as the only copy of MpcA was as resistant to DOC as the wild-type

strain, suggesting that MpcA-Strep is functional. Among all the proteins in the proteome, we specifically identified three proteins, MpcE, MpcF and MpcD, as co-precipitating with MpcA. (Fig. 8). The data demonstrate that MpcEFDA is capable of forming a complex that exhibits a strong protein-protein interaction since the pulldown was generated in the absence of cross-linking agents. We proposed that this complex spans the periplasm, as evidenced by the homology of the ABC transporter in IM to the MlaEFDB complex, with a needle prediction structure of MpcD similar to the C-terminal domain of PqiB and the OM lipoprotein MpcA being homologous to the PqiC OM lipoprotein (Appendix Fig. S7).

## Discussion

The present study characterises a newly identified complex potentially spanning the periplasmic space, namely the Mpc (Mla-Pqi chimeric) system, connecting IM to OM in *B. abortus*. All mutants generated in this system were unable to grow in the presence of DOC (Fig. 2), a compound known to affect bacterial membrane integrity (Urdaneta and Casadesus, 2017). This finding suggests that the Mpc system may play a crucial role in survival under conditions of envelope stress, such as those induced by antimicrobial peptides generated by immune cells (Sawyer et al, 1988). Conversely, it is well-established that wild-type *Brucella* strains exhibit high resistance to cationic peptides (Martínez de

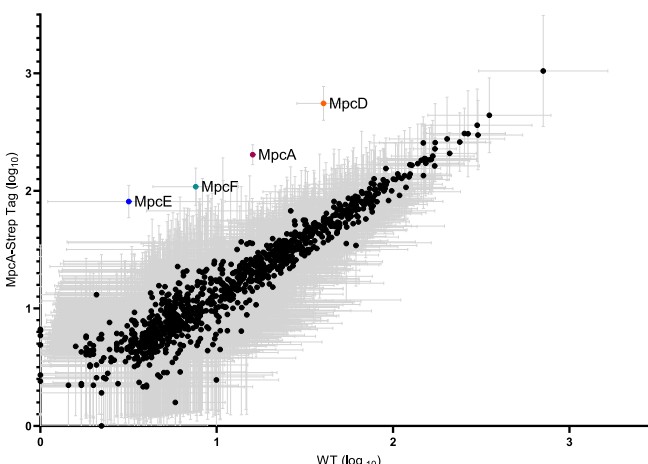

**Figure 8. MpcEFDA is a stable complex.**

Proteins co-purified with MpcA-Strep Tag were identified by MS. The data represent the mean ± SD of total spectrum count in log$_{10}$ for each identified protein (1508) from three biological replicates. The Total Spectrum Count obtained from the pull-down assay on MpcA-Strep Tag is related to the pull-down assay on WT (negative control).

Tejada et al, 1995). Consequently, it would be worthwhile to test the percentage of cell viability after exposure to different antimicrobial peptides, as previously performed in the sensitivity assay (Martínez de Tejada et al, 1995). This suggests that these bacteria encounter cationic peptides during the natural infection, or other envelope stress conditions unidentified so far. When comparing the survival of the different mutants at 2 h post-infection (Fig. 3A), the Δ*mpc* operon seems to be less affected compared to the individual mutants. This could be potentially explained by a toxicity generated by the presence of an incomplete Mpc system, which would worsen the survival defect in the early phase of the infection. Given the inability of the *mpc* mutants to survive the initial stages of infection in macrophages (Fig. 3), we propose that *B. abortus* requires this system to survive attack on its cell envelope at the beginning of its intracellular trafficking. At this stage of the trafficking process, *B. abortus* resides within endosomal *Brucella*-containing vacuoles (eBCVs), where bactericidal conditions are particularly important (Celli et al, 2003; Porte et al, 1999). These eBCVs are characterised by an acidic pH (Porte et al, 1999) and the presence of markers of late endosomes, including LAMP1 and Rab7 (Celli et al, 2003; Starr et al, 2008). Furthermore, it has been demonstrated that *B. abortus* growth is arrested in these compartments (Deghelt et al, 2014), which may be necessary to enable survival in the presence of adverse conditions such as starvation, acidic pH and envelope stress encountered in eBCVs. The necessity of the Mpc system in other infection models is corroborated by the attenuation of the *mpc* genes in Tn-seq analyses conducted on macrophages and mice infection models (Potemberg et al, 2022; Sternon et al, 2018). This result corroborated the initial function attributed to MCE protein in *M. tuberculosis*, which was that of a mediator of mammalian cell entry (Arruda et al, 1993). Indeed, the cell entry defect could be characterised as a survival decrease at the beginning of intracellular trafficking. The sensitivity of the *mpc* mutants to the infection conditions correlates with an alteration in the composition of the

lipids present in the OMVs. Given the homology of the Mpc system to the Mla and Pqi systems (Appendix Figs. S4–S7), which are involved in the transport of lipids between the IM and OM (Coudray et al, 2020; Malinverni and Silhavy, 2009; Nakayama and Zhang-Akiyama, 2017; Tang et al, 2021), it is very likely that the Mpc system mediates the exchange of lipids between IM and OM in *B. abortus*. If the lipid composition of the OMVs correlates with the lipid composition of the OM, which is currently unknown for *B. abortus*, the absence of the Mpc system could alter the lipid composition of the OM, which would be a crucial factor in enabling *B. abortus* to respond effectively to adverse conditions in eBCVs. Furthermore, alterations of the OM composition may also provide an explanation for the recurrent isolation of *asmA*-like mutants in Tn-seq screens with *B. abortus* or *Brucella melitensis* in macrophages and mouse models of infection (Potemberg et al, 2022; Sternon et al, 2018). It is noteworthy that an *asmA* mutant has also been isolated in the DOC sensitivity Tn-seq (Fig. 1), which is potentially coding for a 'short' version of AsmA, while a longer AsmA and a TamB homologue are also predicted by homology analysis of the *B. abortus* genome. This DOC-sensitivity of an *asmA* mutant indicates that the AsmA-like protein could have a role in maintaining the integrity of the OM. This is consistent with the proposed AsmA-like function of anterograde diffusive flow of phospholipids from the IM to the OM (Douglass et al, 2022; Grimm et al, 2020; Kumar and Ruiz, 2023) being conserved in *B. abortus*. In contrast to *V. parvula* (Grasekamp et al, 2023), the double deletion of *asmA* in the *mpc*-operon mutant did not partially rescue the sensitivity to DOC (Fig. EV4A,B), suggesting that AsmA and Mpc may be involved in similar but non-redundant physiological functions related to lipid transport. Collectively, these results indicate that the composition of the OM is a critical determinant of host-pathogen interactions. Further investigations are necessary to gain a deeper understanding of OM properties, including its composition, structure, biogenesis and its role in responding to various stresses encountered during the infectious cycle.

Our results, as illustrated in Fig. 4, indicate a decrease in the proportion of CL in OMVs and a general decrease in OL and LOL in *mpc* mutants. With this result, it is not possible to determine the direction of lipid transport. Indeed, the absence of CL in OMVs in the *mpc* mutants could be attributed to either the lack of anterograde (IM to OM) CL transport or to the accumulation of other PLs in OM in the absence of retrograde (OM to IM) transport of PLs, which would dilute CL in the OM. In light of the observed defect in OM integrity in the *mpc* mutants, it can be postulated that the observed decrease in OL and LOL may result from a rescue phenomenon. Indeed, if the PL transport is imbalanced in the absence of the Mpc system, it is plausible that the bacteria adjust the pools of PLs to compensate for the absence of one of the PLs transporters, resulting in a decrease in the relative concentration of OL and LOL in the envelope. Moreover, our results demonstrate that the observed sensitivity phenotype is effectively induced by the absence of the Mpc system, rather than by the lower abundance of CL, OL or LOL in the OM (Fig. 4). The phenotype of the *cls* mutant, which encodes a non-redundant CL synthase in the tested condition (Fig. 5A), is similar to the wild-type strain in terms of DOC sensitivity and survival at short times post-infection (Fig. 5B,C) (Palacios-Chaves et al, 2011). The *olsA* and *olsB* deletion mutants, which encode ornithine lipid biosynthesis,

exhibited identical results to the WT in terms of DOC sensitivity and survival at short times post-infection, in accordance with previous observations (Palacios-Chaves et al, 2011). However, the deletion of *olsA* in the *mpc* operon background allows a restoration of the wild-type phenotype (Fig. 5B,C). This suggests that the accumulation of LOL (Gao et al, 2004) may have an impact on the DOC sensitivity and survival at short times post-infection only when the Mpc system is absent. It has been demonstrated that the accumulation of lyso-PLs is involved in the stress resistance and virulence of bacteria (Zheng et al, 2017). We cannot exclude that other modifications to the lipid composition of the membranes may also affect the survival rate at short times post-infection. Collectively, these findings indicate that the absence of the Mpc system may have a significant impact on the lipid composition of membranes, due to a defect in lipid transport.

The proposed structural model of Mpc forms a channel between IM and OM (Fig. 7). The length around ~250 Å of the complex, from the N-terminus of MpcA to MpcE, is consistent with the measure of the periplasm width evaluated from Cryo-EM pictures (Godessart et al, 2021). The most notable aspect of the predicted structure is the interaction between the MpcA lipoprotein and the hexameric coiled-coil of MpcD, the latter forming an entirely hydrophobic channel. We propose that the octameric form of MpcA engages in a cap-like interaction with MpcD, similarly to the proposed role of PqiC in anchoring the PqiB periplasmic needle and limiting its flexibility (Cooper et al, 2024). This hypothesis is corroborated by the results of the pull-down assay (Fig. 6), which demonstrated that Mpc interactions are strong and form a stable complex. At the evolutionary level, it is not excluded that Mpc systems may have been ancestral, leading to diversification in different systems (Mla and Pqi). It can be hypothesised that intermembrane complexes with various architectures based on one or several MCE domains and a hexameric alpha-helical coiled-coil are relatively ancient. This proposition is substantiated by the cryo-EM structure of the Mce1 heterohexamer, which forms an elongated needle structure that spans the analogous space of the periplasm in diderm bacteria, with a globular domain on the C-terminal end lying proximal to the mycobacterial OM (Chen et al, 2023a). In addition, the predicted structure of *V. parvula* exhibited a comparable configuration, but with an inserted beta barrel within the OM (Fig. 7) (Grasekamp et al, 2023). One common feature of Mla, TGD and Mpc systems is the presence of a complex homologous to ABC transporters in the IM, suggesting that it could be an ancestral trait in the evolution of these systems. It is particularly noteworthy that the TGD complex, which has a similar needle structure, located within the chloroplast envelope, plays a pivotal role in the process of lipid trafficking (Roston et al, 2012). The complete hydrophobic nature of the channel interior formed by hexameric MpcD is consistent with the channelling of acyl chains of the PLs. However, PLs are amphipathic molecules since they have a charged polar group. We hypothesise that PLs may be transported with counter ions to neutralise the charge of the head group, or two head groups if they are transported by two. The charged polar group could also be inserted between two alpha helices in a hydrophilic environment. The amphipathic nature of the PLs must be accommodated in such a hydrophobic tunnel, a process that will be elucidated by future investigation. Homologues of the Mpc system, with a predicted structure of an elongated alpha-helical C-terminal domain for MpcD, are also found in many Hyphomicrobiales, but with a large variation in the length of the channel (Fig. EV2; Appendix Fig. S4C). This large variation may reflect different periplasmic widths or different angles of the MpcD relative to the IM and OM. It can therefore be surmised that the Mpc system described here is a conserved system, at least in the Hyphomicrobiales, whose function would require further investigation in several other bacteria of interest for basic knowledge, health and biotechnology.

# Methods

**Reagents and tools table**

| Reagent/resource | Reference or source | Identifier or catalogue number |
|---|---|---|
| **Experimental Models** | | |
| *Escherichia coli* DH10B | Invitrogen | |
| *Escherichia coli* S17-1 | Simon et al, 1983 | |
| *Escherichia coli* MFD*pir* | Ferrières et al, 2010 | |
| *Brucella Abortus 544, Nal*<sup>R</sup> | J-M. Verger, INRA, Tours | |
| **Recombinant DNA** | | |
| pNPTS138 | M. R. K. Alley, Imperial College of Science, London, UK | |
| pBBR2 MCS | Deghelt et al, 2014, Unamur, BE | |
| pMR10 | C.D. Mohr and R.C. Roberts, Stanford University, US | |
| pXMCS-2 mini-Tn5 Kan<sup>r</sup> | Sternon et al, 2018, Unamur, BE | |
| pJQ200 Δ*olsA* | Palacios-Chaves et al, 2011, Universidad de Navarra, Spain | |
| pJQ200 Δ*olsB* | Palacios-Chaves et al, 2011, Universidad de Navarra, Spain | |
| pBBR2 MCS *asmA* | G. Potemberg, PhD thesis, Unamur, BE | |
| **Antibodies** | | |
| anti-lipopolysaccharide A76-12G12 monoclonal antibody | Cloeckaert et al, 1993 | |
| goat anti-IgG mouse Alexa 514 | Sigma Aldrich | Cat # A31555 |
| anti-IgG rabbit Pacific blue | Sigma Aldrich | Cat # P10994 |
| **Oligonucleotides and other sequence-based reagents** | | |
| PCR primers | This study | Appendix Table S3 |
| **Chemicals, enzymes and other reagents** | | |
| Tryptic Soy Broth | Difco™ | 211825 |
| kanamycin | PanReac AppliChem | A1493,0025-ITW |
| nalidixic acid | ThermoFisher | B25096 |
| gentamicin | Sigma | G1397 |
| ampicillin | Sigma | A9518 |

| Reagent/resource | Reference or source | Identifier or catalogue number |
|---|---|---|
| Q5® High-Fidelity DNA Polymerase | New England Biolabs | A9518 |
| GoTaq® DNA polymerase | Promega | M7848 |
| T4 DNA ligase | Invitrogen | 100004917 |
| EcoRV | New England Biolabs | R3195S |
| Sodium deoxycholate | VWR | 0613-50 g |
| Proteinase K | Merck | 740506 |
| Dulbecco's Modified Eagle Medium (DMEM), GlutaMAX™ | Gibco | 61965-026 |
| foetal bovine serum (FBS) | Gibco | 10270-106 |
| Triton X-100 | VWR | 0694-1 L |
| phosphate-buffered saline (PBS) | VWR | 392-0442 |
| paraformaldehyde (PFA) | Thermo Fisher | 047392.9 M |
| Bovine serum albumin (BSA) | VWR | 422361 V |
| Fluoromount-G™ Mounting Medium | Invitrogen | 422361 V |
| CellLytic™ B cell lysis reagent | Sigma | C8740 |
| Ready-Lyse lysozyme | Thermo Fisher | R1810M |
| DNAse I | Sigma | 10104159001 |
| cOmplete™ EDTA free protease inhibitor cocktail | Roche | 11873580001 |
| MagStrep "type 3" XT bead | Iba Lifesciences | 2-5090-010 |
| **Software** | | |
| PyMOL Version 2.4.1 | https://pymol.org | |
| GraphPad Prism 8.4.3 | https://www.graphpad.com | |
| Scaffold 5.2.2 | https://www.proteomesoftware.com | |
| FIJI v.2.1.0 | https://imagej.net | |
| Agilent MassHunter Profinder (v. 10.0) | https://www.agilent.com | |
| Domain Enhanced Lookup Time Accelerated BLAST | https://blast.ncbi.nlm.nih.gov/Blast.cgi?PAGE=Proteins | Boratyn et al, 2012 |
| NetSurfP-2.0 | https://services.healthtech.dtu.dk/services/NetSurfP-2.0/ | Klausen et al, 2019 |
| Mascot 2.8.1 | https://www.matrixscience.com/ | |
| **Other** | | |
| Illumina NextSeq HiSeq | Illumina | Fasteris company, Geneva, Switzerland |
| Nikon Eclipse Ti2 equipped with a phase-contrast objective Plan Apo λ DM100XK 1.45/ 0.13 PH3 and a Hamamatsu C13440-20CU ORCA-FLASH 4.0. | Fluorescence microscopy | |

| Reagent/resource | Reference or source | Identifier or catalogue number |
|---|---|---|
| quadrupole-time-of-flight mass spectrometry (LC-MS (QTOF)) | Lipidomics mass spectrometry | Fernández-García et al, 2023 |
| nano-LC-ESI-MS/MS tims TOF Pro | Bruker | |

## Bacterial strains and media

*Brucella abortus* 544 Nal[R] strain (referred to as the WT in this study) and its derivatives were cultivated in TSB-rich medium (Difco™ Tryptic Soy Broth) at 37 °C. All *Escherichia coli* strains were grown in LB Broth Base (Lennox formulation) at 37 °C. All the strains used in this study are listed in Appendix Table S1. Antibiotics were added, when it is necessary, at the following concentrations: kanamycin (10 or 50 μg/mL for genomic or plasmidic resistance, respectively), nalidixic acid (25 μg/mL), gentamicin (20 μg/mL) and ampicillin (100 μg/mL).

## Strains construction

All bacterial strains, plasmids, open reading frames and primers used in this study are listed in the Appendix Tables S1–4, respectively. For deletion strains, ~500 bp upstream and downstream of the coding sequence of interest were amplified from the purified *B. abortus* 544 gDNA by PCR using Q5® High-Fidelity DNA Polymerase (New England Biolabs). The fragments were assembled by overlapping PCR. For the MpcA-StrepTag strain, 500 bp upstream and downstream of the C-terminal region of the *mpcA* gene and the StrepTag sequence (AWSHPQFEK) with a linker (GGGSGGS) were amplified by PCR. The fragments were assembled by overlapping PCR. For deletion strains, the deletion alleles were devoid of an antibiotic resistance cassette, to obtain markerless mutant strains. For complemented strains, the genes of interest were amplified. The resulting amplicons were purified from agarose gels and digested as the destination vector with the corresponding restriction enzymes (New England Biolabs) and ligated (T4 DNA ligase, Promega) overnight at 20 °C. The ligation products were transformed in *E. coli* DH10B, and clones were screened by PCR using GoTaq® DNA polymerase (Promega). Selected plasmids were checked by sequencing and transformed into *E. coli* S17-1 or MFDpir strain to allow conjugation to the *B. abortus* 544 Nal[R] strain. Fusion and deletion strains were made by allelic exchange on the chromosome using a non-replicative plasmid, pNPTS138 (Deghelt et al, 2014). Complemented strains were made using a replicative plasmid (pBBR2 MCS or pMR10), and the genes of interest were under the control of the *E. coli lacZ* promoter or their endogenous promoter, respectively.

## Transposon sequencing assay

One millilitre of an overnight culture of *B. abortus* 544 Nal[R] strain was mixed with 50 μL of an overnight culture of the conjugative *E.*

*coli* MFDpir strain carrying the pXMCS-2 mini-Tn5 Kan^R plasmid (Sternon et al, 2018). The plasmid facilitated the straightforward generation of a high number of transposon mutants due to the hyperactive Tn5 transposase encoded by the plasmid. The mating mixture was incubated overnight at room temperature (RT) on TSB agar plates supplemented with *meso*-2,6-diaminopimelic acid (300 μM). The resulting *B. abortus* mini-Tn5 libraries were selected on TSB agar plates supplemented with kanamycin. The mutant library with envelope stress condition, the medium was supplemented with sterile 0.015% sodium deoxycholate (w/v). This was the higher concentration that did not alter the growth of the wild-type strain on TSB plates. Mini-Tn5 mutagenesis generates insertion of the transposon at only one locus per genome, as demonstrated previously for *Brucella* (Lestrate et al, 2000; Sternon et al, 2018).

## Analysis of essential genes for growth

Genomic DNA was extracted from each transposon library using standard techniques and prepared for sequencing of the mini-transposon library. *B. abortus* Tn mutants from each plate were collected, mixed in 2% SDS, and killed by heating (1 h, 80 °C). The lysate was incubated for 3 days at 37 °C under constant agitation in TENa buffer (pH = 8.0, Tris 50 mM, EDTA 50 mM, 100 mM NaCl), which was complemented with 0.1 mg/mL Proteinase K (Merck) and 0.5% SDS. Subsequently, an equal volume of isopropanol was added to precipitate the gDNA, which was then washed with 70% ethanol. The gDNA was resuspended in deionised water, and the regions flanking the mini-Tn5 were sequenced (Fasteris company, Geneva, Switzerland). In order to map the insertion sites of mini-Tn5, the libraries were sequenced on an Illumina HiSeq with a primer hybridised at the border of the transposon, with its 3' end pointing towards the flanking gDNA. Raw reads were filtered using the cutadapt algorithm and then mapped on the *B. abortus* 544 genome using the Burrows-Wheeler alignment (BWA) algorithm (114,039,048 mapped reads in Control condition and 134,058,965 mapped reads in the DOC condition). The read counts were determined using samtools (Coppine et al, 2020).

In the control condition, there was an average of one unique insertion site every 2.62 bp, and in the envelope stress condition, every 2.48 bp, saturating the *B. abortus* genome in both conditions. The mapping analysis (Coppine et al, 2020) yielded an insertion number per bp for 80% of each open reading frame (ORF—excluding the first and last 10% of the ORF). Essential genes were identified based on the insertion number per bp. An open reading frame (ORF) is considered essential if the insertion number per bp is below 0.1, which corresponds to one insertion every 10 bp (Fig. EV1). Out of the predicted 3390 ORFs in the *B. abortus* genome, 551 have been identified as essential for growth in rich medium, which corresponds to 16.2% of the predicted genes. To enable a quantitative analysis of each ORF in different conditions, we calculated a transposon insertion frequency (TnIF) parameter. This frequency corresponds to the logarithm in base 10 of the mapped read numbers (*ES.Reads*) for the central 80% of the ORF [$\log_{10}(ES.Reads + 1)$]. To identify genes required for growth in the presence of DOC, when the envelope is destabilised, we compared the TnIF of the stress condition to the control condition to obtain a ΔTnIF for each ORF (ΔTnIF = TnIF$_{DOC}$ - TnIF$_{ctrl}$). Thus, ORF with

a negative ΔTnIF value correspond to mutants with a DOC-sensitivity phenotype (Fig. 1; Appendix Table S5).

## Envelope stress growth tests

Overnight culture of *B. abortus* 544 and derivative strains are normalised at OD$_{600}$ 0.1 and diluted in tenfold increments. Fifteen μL of each dilution are spotted on a TSA plate containing or not sodium deoxycholate at 0.005% (w/v).

## J774A.1 macrophage culture and infection

J774A.1 (ATCC) macrophages were cultured in Dulbecco's Modified Eagle Medium (DMEM), GlutaMAX™ supplement medium (Gibco) supplemented with 10% heat-inactivated foetal bovine serum (FBS, Gibco) at 37 °C in a 5% CO$_2$ atmosphere. The day before infection, cells were inoculated at 10$^5$ cells/mL in 24-well plates (0.5 mL per well). Overnight culture of *Brucella* was prepared in DMEM at a multiplicity of infection (MOI) of 50 bacteria per cell. The suspension of bacteria was placed on the cell and centrifuged at 169 rcf for 10 min at 4 °C. Infected cells were incubated for 1 h at 37 °C in a 5% CO$_2$ atmosphere. Subsequently, the culture medium was removed and replaced with fresh medium supplemented with 50 μg/mL gentamicin to kill extracellular bacteria. After 1 h, the medium was changed to DMEM with 10% FBS with 10 μg/mL gentamicin.

## CFU counting

At 2, 5, 24 and 48 h post-infection, cells were washed twice with phosphate-buffered saline (PBS), and macrophages were lysed using PBS–0.1% Triton X-100 for 10 min at 37 °C and flushed to recover bacteria. Twenty μL of dilutions bacteria were plated onto a TSB agar plate, and the colony-forming unit (CFU) were counted after incubation at 37 °C.

## Immunolabeling of infected host cells

Labelling of intra- and extra-bacteria in host cell, was carried out as previously reported (Deghelt et al, 2014) with minor modifications. Briefly, the cells were inoculated at 3 × 10$^5$ cells/mL in a 24-well plate with coverslips. At 2 h post-infection at MOI 1000, cells on the coverslips, were washed three times with PBS and fixed with 4% paraformaldehyde (PFA, Thermo Fisher) at RT for 30 min. Coverslips were washed twice with PBS, and first labelled for 45 min at RT with a homemade rabbit serum against heat-killed *Brucella*, diluted 2000 times in PBS with 3% BSA. Subsequently the permeabilization with 0.1% Triton X-100 in 3% BSA for 10 min, the coverslips were incubated for 45 min at RT with anti-lipopolysaccharide A76-12G12 monoclonal antibody (Cloeckaert et al, 1993) undiluted in 3% BSA, washed three times with PBS and incubated with secondary antibodies solution (each diluted 500 times) containing a goat anti-IgG mouse Alexa 514 (Sigma Aldrich) and an anti-IgG rabbit Pacific blue (Sigma Aldrich), in PBS with 3% BSA and 0.1% Triton X-100, for 30 min at RT. The coverslips were washed three times with PBS, then once with demineralised water and mounted with Fluoromount-G™ Mounting Medium (Invitrogen).

## Microscopy and analysis

*Brucella* strains and infected J774A.1 cells were observed with a Nikon Eclipse Ti2 equipped with a phase-contrast objective Plan Apo λ DM100XK 1.45/0.13 PH3 and a Hamamatsu C13440-20CU ORCA-FLASH 4.0. Fluorescence images were analysed using FIJI v.2.1.0, a distribution of ImageJ (Schindelin et al, 2012). Lookup tables (LUT) were adjusted to the best signal–to–background ratio.

## Isolation of *Brucella* OMVs

Six 48-h cultures of 800 mL were inactivated with 0.5% phenol and centrifuged at 8200 rcf at RT for 20 min at 4 °C. The pellet was conserved, and the supernatant was concentrated using the Pellicon® tangential flow filtration system, with a membrane of 100,000 Dalton exclusion limit. The concentrated supernatant was centrifuged at 8200 rcf, 4 °C for 20 min to discard the sediment. Then, to aggregate the OMVs, the supernatant was frozen at −20 °C. The defrosted supernatant was ultra-centrifuged at 47,000 rcf, 4 °C for 3 h. The pellet, the isolated OMVs, was resuspended in deionised water and dialysed at 4 °C with deionised water for 3 days and finally frozen at −80 °C. Both the culture pellet and the isolated OMVs suspension were lyophilised by freeze dryer (TELSTAR CRYODOS 50).

## Lipidome analysis

Total lipids were extracted as described by Bligh and Dyer (Bligh and Dyer, 1959; Daniels et al, 1993). Briefly, 500 μL of resuspended bacteria or OMVs (20 mg of lyophilised samples) in deionised water were vortexed for 3 h with 650 μL of chloroform and 1300 μL of methanol. About 650 μL each of chloroform and deionised water were added and mixed for 30 min. After short centrifugation to separate aqueous and organic phases, the organic phase was kept and evaporated with a flow of nitrogen.

Thin layer chromatography (TLC) was then performed on the lipid extractions to separate the phospholipids into different bands (Ando and Saito, 1987). The lipid extracts were analysed on silica gel 60 high-performance thin layer chromatography plates (Merck Chemicals) and chromatography was performed in a solution composed of chloroform (TCM), methanol (MeOH) and demineralised water ($H_2O_{dd}$)(14:6:1, respectively v:v). or TCM, MeOH and acid acetic glacial (AcOH) (13:5:2, respectively v:v). Plates were developed with a solution of $CuSO_4$ (10%, w/v) in acid phosphoric (8%, v/v) at 180 °C. The lipids were separated into four bands (1) phosphatidylcholine (PC), (2) phosphatidylethanolamine and ornithine lipid (PE/OL), (3) phosphatidylglycerol (PG) and (4) cardiolipin (CL). The intensity of each band was measured with the GS-800™ densitometer (Bio-Rad) and Quantity One—4.6.6 software to enable comparison between samples.

Complementary to TLC analysis, a mass spectrometry-based lipidomic analysis was performed using ultra high-performance liquid chromatography coupled to quadrupole-time-of-flight mass spectrometry (LC-MS (QTOF)) as previously described (Fernández-García et al, 2023), with minor modifications. Briefly, lipid residues obtained by Bligh and Dyer for whole bacteria or OMVs were resuspended in 300 μL of methanol:chloroform containing 25 mg/L of d17:0 sphinganine (IS1). Reextraction of lipids was performed by vortexing at room temperature RT for 20 min. Lipid extracts were transferred to individual LC-MS vials, where 20 μL of SPLASH Lipidomix® lipid standard mixture (IS2, Avanti Polar Lipids, CA, USA) were added to each vial. Then, extracts were evaporated to dryness using a vacuum concentrator at 37 °C. Residues were reextracted in 100 μL of methanol:chloroform (2:1, v/v) by thorough vortexing for 30 min at RT. Sample analysis was performed using an Agilent 1290 HPLC system coupled to an Agilent 6545 QTOF mass analyser (both from Agilent Technologies, Santa Clara, CA, USA), equipped with an ESI ionisation source. Two run sequences of sample extracts corresponding to positive (ESI+) and negative (ESI−) ionisation modes were performed. 2 (ESI+) or 5 μL (ESI−) were injected into an Agilent InfinityLab Poroshell 120 EC-C18 column (3.0 mm × 100 mm, 2.7 μm; Agilent Technologies, Santa Clara, CA, USA) equipped with an Agilent InfinityLab Poroshell 120 EC-C18 guard column (3.0 mm × 5 mm, 2.7 μm; Agilent Technologies, Santa Clara, CA, USA), operating at 50 °C. A mobile phase flow rate of 0.6 mL/min was maintained throughout the chromatographic gradient. Firstly, 70% B was held for 1 min. Secondly, 86% B was achieved at 3.5 min and held until 10 min. Next, 100% B was achieved at 11 min and held until 17 min, followed by 2 min of re-equilibration time at 70% B. The total method runtime was 19 min. Metabolites were ionised using an ESI source with a nebuliser at 50 psi, a drying gas temperature of 200 °C, a drying gas flow rate of 10 L/min, a sheath gas temperature of 300 °C, and a sheath gas flow rate of 12 L/min. In both polarity modes (ESI+ and ESI−), the capillary, fragmentor, skimmer, and octupole radiofrequency voltages were set to 3500, 150, 65 and 750 V, respectively. Firstly, full MS was selected as the data acquisition mode at an acquisition rate of 3.5 spectra/s over a mass range of m/z 50 to 3000 for ESI−. Next, two iterative auto MS/MS analyses (under both polarity modes) were conducted under identical chromatographic conditions, to provide MS/MS data allowing high-confidence lipid annotation. MS/MS spectra were systematically acquired via iterative MS/MS mode over subsequent injections from a sample pool. Collision energy was set to 20 eV for ESI+, and 40 eV for ESI−. Lipids were annotated by using Agilent MassHunter Lipid Annotator (v. 1.0) followed by manual curation of annotations using Agilent MassHunter Qualitative Analysis (v. 10.0) considering lipid subclass diagnostic ions, adduct pattern, and elution order (Köfeler et al, 2021). OL and LOL were manually annotated using Agilent MassHunter Qualitative Analysis in accordance with reported fragmentation spectra (Zhang et al, 2011). The curated annotation data matrix was used as input for targeted compound integration using Agilent MassHunter Profinder (v. 10.0).

## Homology and structure predictions

Mpc proteins of *B. abortus* and *E. coli* Mla proteins were searched using standard protein BLAST with the DELTA-BLAST (Domain Enhanced Lookup Time Accelerated BLAST) (Appendix Fig. S4) (Boratyn et al, 2012). Secondary structures of homologues of MpcD in Hyphomicrobiales were predicted with NetSurfP-2.0 (Fig. EV2) (Klausen et al, 2019). The interaction between Mpc proteins, $MpcE_2F_2D_6$ and $MpcD_6A_8$, were predicted using AlphaFold3 multimer predictions of biomolecular interactions (Appendix Fig. S6) (Abramson et al, 2024). Resulting predictions were visualised and analysed in The PyMOL Molecular Graphics System, Version 2.4.1, Schrödinger, LLC.

## Pull-down assay

*B. abortus* cultures of 200 mL (0.8 OD$_{600}$) were harvested by centrifugation at 4500 rcf at 4 °C for 20 min, then washed three times with PBS. The pellet was resuspended in 2 mL of DDM lysis buffer (CellLytic™ B cell lysis reagent (Sigma), 10 mM MgCl$_2$, 0.05% Ready-Lyse lysozyme (v/v), 0.25 mg/mL DNAse I, 0.5% DDM (w/v), cOmplete™ EDTA free protease inhibitor cocktail), incubated 30 min at RT. The lysate was sonicated on ice by applying six bursts of 10 s at 60% amplitude and centrifuged at 12,500 rcf at 4 °C for 15 min. Collect the supernatant and add 1 mg/mL avidin and incubate for 30 min at 4 °C on a wheel, then centrifuge for 5 min at 6000 rcf. The fusion protein MpcA-StrepTag was purified using the MagStrep "type 3" XT bead (Iba) in accordance with the manufacturer's instructions.

The proteins recovered in the elution are treated using Filter-aided sample preparation (FASP) to generate tryptic peptides. About 40 μg of protein adjusted in 150 μl of buffer UA (8 M urea in 0.1 M Tris-HCl pH = 8,5) are filtered on Microcon-30 kDa (Millipore, MRCFOR030), previously rinsed with formic acid 1% (v/v), at 19,800 rcf for 15 min. The filter was then washed three times with 150 μl of Buffer UA (15 min at 19,800 rcf). The proteins were reduced with 100 μL of dithiothreitol (DTT, 0.008 M in UA), incubated on a thermomixer at 24 °C for 15 min, and centrifuged for 10 min at 19,800 rcf. The filter was further washed with 100 μl of Buffer UA (15 min at 19,800 rcf). Then, proteins were alkylated with 100 μL of iodoacetamide (IAA, 0.05 M iodoacetamide in UA), incubated for 20 min at 24 °C in the dark before centrifugation for 10 min at 19,800 rcf. The filter was washed with 100 μl of Buffer UA (15 min at 19,800 rcf). The IAA was quenched by the addition of 100 μl of DTT. After an incubation of 15 min at 24 °C, the column was centrifuged for 10 min at 19,800 rcf, and further rinsed with 100 μl of Buffer UA. The filter was washed three times with 100 μl of Buffer ABC (0.05 M NH$_4$HCO$_3$ in water) and centrifuged for 10 min at 19,800 rcf. Finally, proteins were digested by trypsin (1 μg, Promega) overnight at 24 °C. The digested proteins were recovered by placing the Microcon column on a protein LoBind tube of 1.5 ml, followed by centrifugation for 10 min at 19,800 rcf. The filter was rinsed with 40 μl of Buffer ABC and centrifuged for 10 min at 19,800 rcf. The filtrate was then acidified with a solution of 10% Trifluoroacetic acid (TFA) to reach a 0.2% TFA final concentration. The sample was finally dried in a speed-vac up to 20 μl before mass spectrometry analysis (nano-LC-ESI-MS/MS tims TOF Pro, Bruker) coupled with a nanoUHPLC (nanoElute, Bruker).

Peptides were separated by nanoUHPLC on a 75 μm ID, 25 cm C18 column with an integrated CaptiveSpray insert (Aurora, Ionopticks, Melbourne) at a flow rate of 200 nL/min, at 50 °C. LC mobile phases A was water with 0.1% formic acid (v/v), and B was ACN with formic acid 0.1% (v/v). Samples were loaded directly on the analytical column at a constant pressure of 800 bar. The digest (1 μl) was injected, and the organic content of the mobile phase was increased linearly from 2% B to 15% in 22 min, from 15% B to 35% in 38 min and from 35% B to 85% in 3 min. Data acquisition on the tims TOF Pro was performed using Hystar 6.1 and timsControl 2.0. tims TOF Pro data were acquired using 160 ms TIMS accumulation time, mobility (1/K0) range from 0.75 to 1.42 Vs/cm². Mass-spectrometric analysis were carried out using the parallel accumulation serial fragmentation (PASEF) acquisition method

(Meier et al, 2018). One MS spectra followed by six PASEF MSMS spectra per total cycle of 1.16 s.

All MS/MS samples were analyzed using Mascot (Matrix Science, London, UK; version 2.8.1). Mascot was set up to search the *Brucella abortus* 2308 proteome (3027 entries) downloaded from UniProt (July 2023) and Contaminants_20190304 database, assuming the digestion enzyme trypsin. Mascot was searched with a fragment ion mass tolerance of 0.050 Da and a parent ion tolerance of 15 PPM. Carbamidomethyl of cysteine was specified in Mascot as a fixed modification. Deamidated of asparagine and glutamine, oxidation of methionine and acetylation of the N-terminus were specified in Mascot as variable modifications. Scaffold (5.1.1, Proteome Software Inc., Portland, OR) was used to validate MS/MS-based peptide and protein identifications. Peptide identifications were accepted if they could be established at greater than 97.0% probability to achieve an FDR less than 1.0% by the Percolator posterior error probability calculation (Kall et al, 2008). Protein identifications were accepted if they could be established at greater than 50.0% probability to achieve an FDR less than 1.0% and contained at least two identified peptides. Protein probabilities were assigned by the Protein Prophet algorithm (Nesvizhskii et al, 2003). Proteins that contained similar peptides and could not be differentiated based on MS/MS analysis alone were grouped to satisfy the principles of parsimony. Proteins sharing significant peptide evidence were grouped into clusters.

## Data availability

The mass spectrometry proteomics data have been deposited in the ProteomeXchange Consortium via the PRIDE partner repository with the Project accession PXD057244 (Project DOI: 10.6019/PXD057244). The data were available at https://www.ebi.ac.uk/pride/archive/projects/PXD057244. The mass spectrometry lipidomics data have been deposited in the Metabolomics Workbench with the Project ID PR002294 at https://doi.org/10.21228/M83R72. The Transposon sequencing data have been deposited to the figshare platform, at this link: https://figshare.com/articles/dataset/Tn-seq_reads_in_control_and_deoxycholate_conditions/29244773.

The source data of this paper are collected in the following database record: biostudies:S-SCDT-10_1038-S44318-025-00511-3.

## Peer review information

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

## Acknowledgements

We thank M. Loperena Barber and R. Condez-Alvarez lab for their help in OMVs isolation and TLC assay; E. Carlier, M. Waroquier, C. Desy and F. Tilquin for their logistic support; C. Roomans and G. Lima Mendez for her help in the initial steps of protein structure prediction; and G. C. Demazy and S. Burteau for the MS sample preparation; P. Cherry and K. Poncin for the knowledge regarding pull-down; E. Barbieux for the experience of the Tn-seq assay; and staff at UNamur (https://www.unamur.be/) for financial and logistical support. We thank the URBM researchers, JeanFrançois Collet (UCLouvain) and Basile Beaud Benyahia (Pasteur Institute, Paris), for stimulating discussions. The LABGeM (CEA/Genoscope and CNRS UMR8030), the France Génomique and French Bioinformatics Institute national infrastructures (funded as part of Investissement d'Avenir programme managed by Agence Nationale pour la Recherche, contracts ANR-10-INBS-09, ANR-11-INBS-0013 and ANR-21-ESRE-0048) are acknowledged for support within the MicroScope annotation platform. AL was supported by a FRIA (FNRS) PhD fellowship. This publication is supported by the Walloon Region as part of the funding for the FRFS-WELBIO strategic axis (X.1512.24). The work was also supported by PDR grants T.0058.20 and T.0068.24 from FRS-FNRS, as well as Concerted Research Action 17/22-087 and 22/27-128 from the Fédération Wallonie-Bruxelles. This research was carried out in the frame of project PID2023-146797OB-C31 financed by MCIN/AEI/10.1303910.13039/501100011033.

## Author contributions

**Adélie Lannoy**: Conceptualisation; Investigation; Writing—original draft; Writing—review and editing. **Alexi Ronneau**: Investigation. **Miguel Fernández García**: Formal analysis; Investigation; Writing—review and editing. **Marc Dieu**: Investigation. **Patricia Renard**: Investigation. **Antonia García Fernández**: Formal analysis; Investigation; Writing—review and editing. **Raquel Condez-Alvarez**: Investigation; Writing—review and editing. **Xavier De Bolle**: Conceptualisation; Writing—original draft; Writing—review and editing.

Source data underlying the figure panels in this paper may have individual authorship assigned. Where available, figure panel/source data authorship is listed in the following database record: biostudies:S-SCDT-10_1038-S44318-025-00511-3.

## Disclosure and competing interests statement

The authors declare no competing interests.

# Expanded View Figures

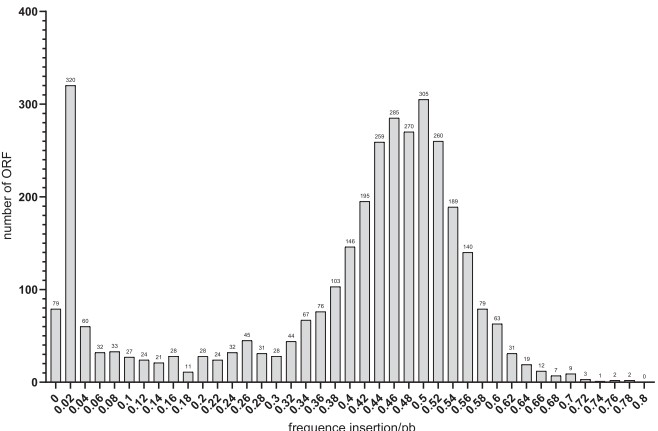

**Figure EV1.  Frequency distribution of the frequency of insertions per pair of bases for mini-Tn5 across all open reading frames (ORFs) in the *B. abortus* genome.**

For each mutated gene, a Transposon insertion frequency (TnIF) is computed. Transposon insertion sites were identified through Illumina sequencing ($n = 1$). The frequency of insertions per pb for each of 3390 ORFs from *B. abortus* genome are represented by classes of 0.02. Under 0.1 frequency of insertions per pb, the ORFs are considered essential for growth on TSA-rich medium, which corresponds to 16.25% of the predicted genes. Source data are available online for this figure.

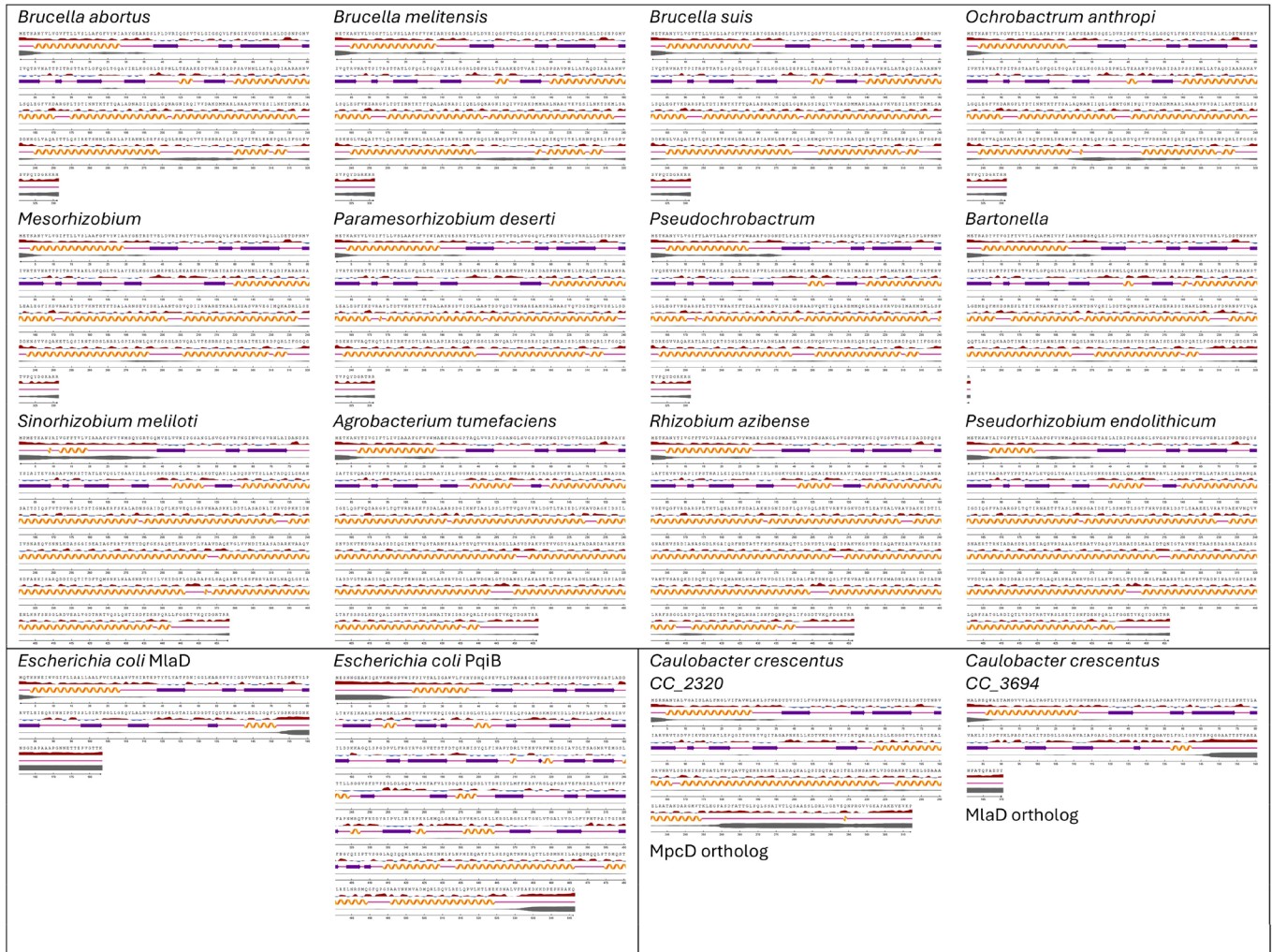

**Figure EV2. Secondary structure prediction by NetSurfP-2.0 (Klausen et al, 2019) of MpcD and homologous proteins in some Hyphomicrobiales bacteria (top panel).**

Secondary structure of MlaD and PqiB in *E. coli* (lower left panel) and secondary structure prediction of proteins in *Caulobacter crescentus* (lower right panel), one homologous to *B. abortus* MpcD and the other homologous to *E. coli* MlaD. Source data are available online for this figure.

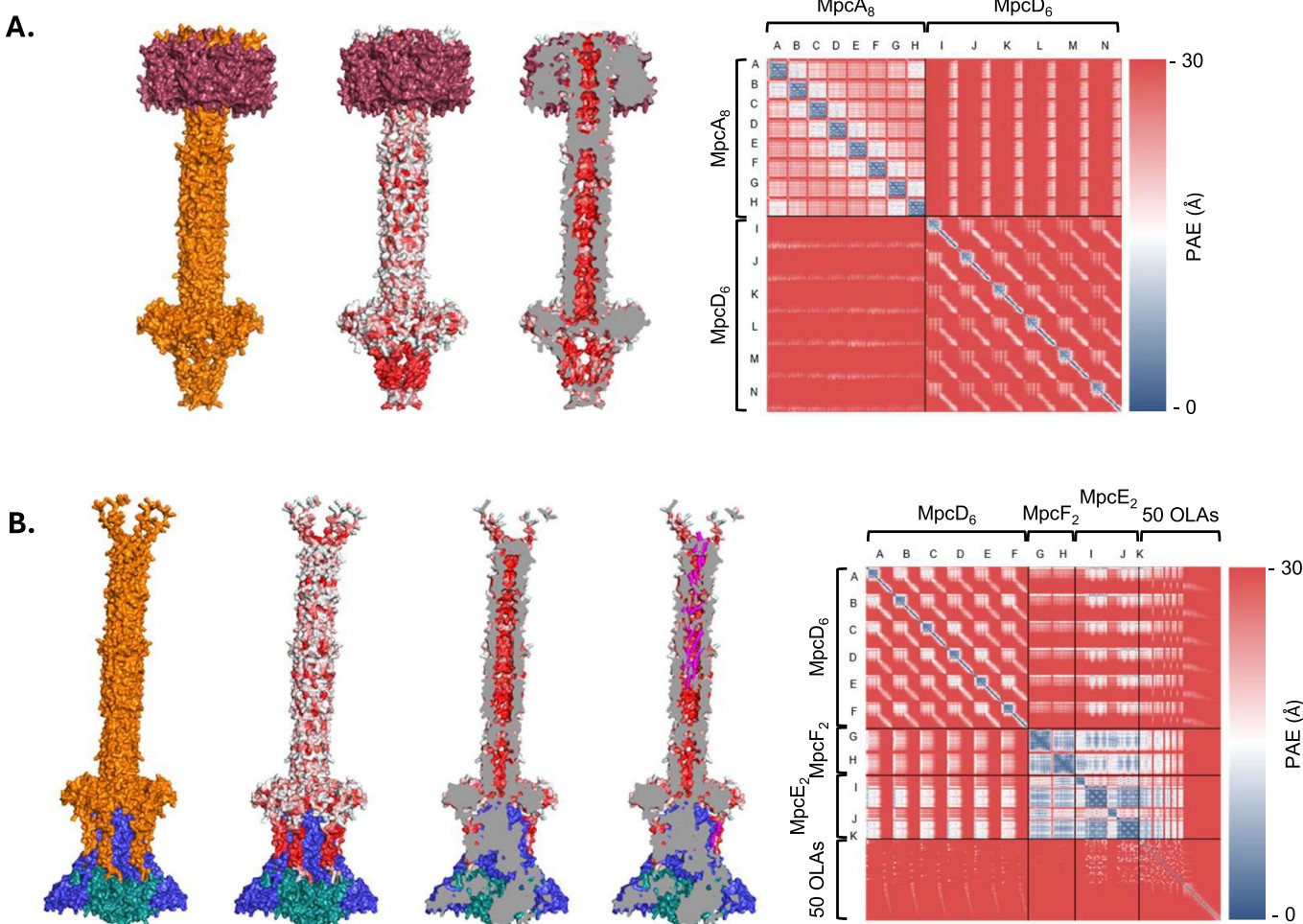

**Figure EV3. The model predictions for MpcD$_6$A$_8$ (top) and MpcE$_2$F$_2$D$_6$ (bottom).**

The surface representation predicted by the AlphaFold3 multimer of the biomolecular interactions of complexes of (**A**) MpcD$_6$A$_8$ (model_1) and (**B**) MpcE$_2$F$_2$D$_6$ (model_1) with the associated predicted aligned error (PAE) maps associated. The first panel shows the predicted interaction with the homodimer of MpcE (blue), the homodimer of MpcF (green), the homo-hexamer of MpcD (orange) and the homo-octamer of MpcA (pink). In the second panel, the MpcD homo-hexamer is coloured according to the Eisenberg hydrophobicity scale using the color_h command in PyMol (hydrophobic regions depicted in red). The third panel shows a slice of the complexes view, allowing observation of the predicted hydrophobicity of the full-length tunnel within the hexameric MpcD channel. The final panel shows the slice of the MpcE$_2$F$_2$D$_6$ complex in predicted interaction with 50 oleic acids (magenta), some of which are situated within the channel formed by the coiled-coil alpha-helices of MpcD.

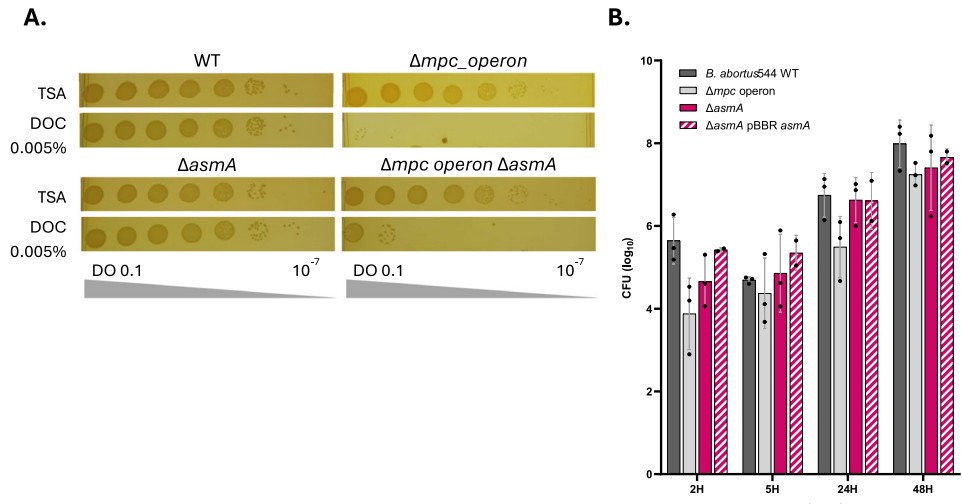

**Figure EV4.   The sensitivity phenotype of *asmA* deletion mutants.**

(A) The plating assay on the *asmA* mutants, the *mpc* operon mutant and double mutants on DOC. All the mutants with deletion of *mpc* operon present the same sensitivity to DOC at 0.005%. The overnight cultures were normalised to 0.1 $OD_{600}$ and serially diluted before being plated onto TSA with or without DOC. (B) Intracellular replication of WT, *mpc* operon, *asmA* mutant mutant and the double mutants were assessed by CFU at 2, 5, 24 and 48 h post-infection of J774.A1 macrophages. The data represents the mean ± SD and were compiled from three independent replicates, represented by the dots. Statistical significance between the results for a given strain and those for the WT was determined using a two-way ANOVA. No significant difference was found ($p \geq 0.05$). Source data are available online for this figure.

