## [Peer Review File · The EMBO Journal]

A chimeric Mla-Pqi lipid transport system is required for *Brucella abortus* survival in macrophages

Adélie Lannoy, Alexi Ronneau, Miguel Fernández García, Marc Dieu, Patricia Renard, Antonia Garcia, Raquel Conde, and Xavier De Bolle

Corresponding author(s): Xavier De Bolle (xavier.debolle@unamur.be)

Review Timeline:

Submission Date:	30th Oct 24
Editorial Decision:	18th Dec 24
Revision Received:	4th Apr 25
Editorial Decision:	30th May 25
Revision Received:	17th Jun 25
Accepted:	25th Jun 25

Editor: Ioannis Papaioannou

Transaction Report:

Dear Prof. De Bolle,

Thank you again for submitting your manuscript EMBOJ-2024-119476 for consideration by The EMBO Journal, and for your patience during peer review. Your manuscript has now been seen by three experts in the field, and we have received the full set of their well-informed and detailed reports, which are included below.

As you will see, all referees recognize that the results presented in this manuscript are interesting and relevant, the provided advance significant, and the manuscript well-written. They also identify, however, a number of limitations and raise several technical and other concerns, which would require both experimental work and rewriting of some sections of the manuscript to be addressed. Among other points, they raise some concerns regarding the identity of the used OMVs in the study, and the extent to which these accurately reflect the lipid composition of the OM. Other concerns relate to over-interpretation of some AlphaFold predictions without experimental validation.

Given the referees' positive and encouraging comments and recommendations, I would like to invite you to submit a thoroughly revised version of the manuscript along with a detailed point-by-point response addressing all referees' comments. I should add that it is The EMBO Journal policy to allow only a single round of major revision, and acceptance of your manuscript will therefore depend on the completeness of your responses in this revised version. Please let me know if you have any questions or comments that you would like to discuss with me.

We generally allow three months as standard revision time (March 17, 2025). As a matter of policy, competing manuscripts published during this period will not negatively impact our assessment of the conceptual advance presented by your study. However, we request that you contact us as soon as possible upon publication of any related work, to discuss how to proceed. Should you foresee a problem in meeting this three-month deadline, please let us know in advance and we may be able to grant an extension.

Thank you for the opportunity to consider your work for publication in The EMBO Journal. I look forward to your revision.

Best regards,

Ioannis

Instructions for preparing your revised manuscript

1. When you are ready to submit the revision, please upload:

- A Word file of the manuscript text (including legends of main Figures, EV Figures and Tables). Please make sure that changes are highlighted (or "tracked") to be clearly visible.

- Individual production-quality figure files (one file per figure). When assembling your figures, please refer to our figure preparation guidelines in order to ensure proper formatting and readability in print as well as on screen:

If the data shown in a figure are obtained from n {less than or equal to} 2, please use scatter plots showing the individual data points.

i. the name of the statistical test used to generate error bars and P values

ii. the number (n) of independent experiments (please specify technical or biological replicates) underlying each data point (discussion of statistical methodology can be reported in the Materials and Methods section, but figure legends should contain a basic description of n , P , and the test applied)

iii. the nature of the bars and error bars (s.d., s.e.m.).

- A point-by-point response to the referees' comments, with a detailed description of the changes made (as a word file). All

referees' concerns must be fully addressed and their suggestions taken on board. When preparing your letter of response to the referees' comments, please bear in mind that this will form part of the Review Process File and will therefore be available online to the community. Please note that you have the possibility to opt out of the transparent process at any stage prior to publication by letting the editorial office know (contact@embojournal.org); if you do opt out, the Review Process File link will point to the following statement: "No Review Process File is available with this article, as the authors have chosen not to make the review process public in this case.". For more details on our Transparent Editorial Process, please visit our website: <https://www.embopress.org/page/journal/14602075/authorguide#transparentprocess>

- Expanded View (EV) files (replacing Supplementary Information) that are collapsible/expandable online. A maximum of 5 EV Figures can be typeset. EV Figures should be cited as "Figure EV1, Figure EV2" etc. in the text, and their respective legends should be included in the manuscript file after the legends of regular figures. See detailed instructions regarding Expanded View files here:

- For the figures that you do NOT wish to display as Expanded View figures, they should be bundled together with their legends in a single PDF file called "Appendix", which should start with a short Table of Contents (including page numbers). Appendix figures should be referred to in the main text as: "Appendix Figure S1, Appendix Figure S2" etc. Please see detailed instructions here: <https://www.embopress.org/page/journal/14602075/authorguide#expandedview>

- A complete author checklist, which you can download from our author guidelines (<https://www.embopress.org/page/journal/14602075/authorguide>). Please note that the checklist will also be part of the Review Process File.

2. Please note that no statistics should be calculated and shown in Figures if $n=2$. Please also note that each p value should be reported as an exact value.

3. Before submitting your revision, primary datasets (and computer code, where appropriate) produced in this study need to be deposited in appropriate public databases (see <https://www.embopress.org/page/journal/14602075/authorguide#dataavailability>).

In particular, we kindly request you to deposit all sequencing and mass spectrometry data produced in your study to appropriate databases. The accession numbers, databases, and the specific URLs (links) should be listed in a formal "Data availability" section (placed after Methods), following the example below:

"The RNA-seq datasets produced in this study are available in the following database:
Gene Expression Omnibus GSE46843 (<https://www.ncbi.nlm.nih.gov/geo/query/acc.cgi?acc=GSE46843>)"

*** All links should resolve to a page where the data can be accessed. ***

*** Please remember to provide in the Data availability section of your revised manuscript reviewer passwords if the datasets are not yet public. ***

*** The Data Availability Section is restricted to new primary data that are part of this study. In case you have no data that require deposition in a public database, please state so instead of referring to the database: "Our study includes no data deposited in public repositories." under the heading "Data availability". ***

4. Please check that the title and the abstract of the manuscript are brief, yet explicit, even to non-specialists. The length of the title should not exceed 100 characters, and the abstract should be a single paragraph not exceeding 175 words.

5. Please also note our reference format: <https://www.embopress.org/page/journal/14602075/authorguide#referencesformat>.

7. Please remember: digital image enhancement is acceptable practice, as long as it accurately represents the original data and conforms to community standards. If a figure has been subjected to significant electronic manipulation, this must be noted in the figure legend or in the "Materials and Methods" section. The editors reserve the right to request original versions of figures and the original images that were used to assemble the figure.

8. Our journal encourages inclusion of data citations in the reference list to directly cite datasets that were obtained from public databases. Data citations in the article text are distinct from normal bibliographical citations and should directly link to the database records from which the data can be accessed. In the main text, data citations are formatted as follows: "Data ref: Smith et al, 2001" or "Data ref: NCBI Sequence Read Archive PRJNA342805, 2017". In the Reference list, data citations must be labeled with "[DATASET]". A data reference must provide the database name, accession number/identifiers, and a resolvable

link to the landing page from which the data can be accessed at the end of the reference. Further instructions are available at: <https://www.embopress.org/page/journal/14602075/authorguide#referencesformat>.

9. We request authors to consider both actual and perceived competing interests. Please review our policy (<https://www.embopress.org/page/journal/14602075/authorguide#conflictsofinterest>) and update your competing interests statement if necessary. Please name this section 'Disclosure and competing interests statement' and place it after the Acknowledgements section.

10. Please note that all corresponding authors are required to provide an ORCID ID upon submission of a revised manuscript (<https://orcid.org/>). Please find instructions on how to link your ORCID ID to your account in our manuscript tracking system in our Author guidelines (<https://www.embopress.org/page/journal/14602075/authorguide#authorshipguidelines>).

11. We use CRediT to specify the contributions of each author in the journal submission system. CRediT replaces the author contribution section, which should be removed from the manuscript. Please use the free text box to provide more detailed descriptions. See also guide to authors: <https://www.embopress.org/page/journal/14602075/authorguide#authorshipguidelines>.

13. We would also welcome the submission of cover suggestions or motifs to be used by our Graphics Illustrator in designing a cover.

14. Please use the link below to submit your revision:
<https://emboj.msubmit.net/cgi-bin/main.plex>

Referee #1:

Despite its critical role in Gram-negative bacteria, the mechanism of phospholipid transport from the inner membrane to the outer membrane remains poorly understood, particularly in species other than *E. coli*. Lannoy et al. identified a potential lipid transport system in *Brucella abortus* using a Tn-Seq approach to screen for detergent-resistant mutants. This so-called Mpc complex appears to be a hybrid of the known Mla and Ppi systems, presumably forming a bridge between the IM and OM. Computational predictions suggest that the Mpc system is a streamlined system comprising only four distinct subunits, and pull-down experiments suggest interactions among these proteins.

This is an interesting and relevant study that has the potential to represent a significant advance in the field. The manuscript is well written; however, my concern is that the conclusions regarding the OM composition are not sufficiently supported by the data (as outlined below). Furthermore, a substantial portion of the proposed model depends heavily on computational predictions.

Major concern:

The authors demonstrate that the absence of the Mpc system alters the lipid composition of OMVs. I was wondering whether OMVs accurately reflect the composition of the OM. My understanding is that OMVs can originate from vastly diverse processes such as blebbing or explosive lysis, and one of their functions is expel "waste" components (LPS, proteins, ...) from the OM. Moreover, since lipids are unevenly distributed within the membrane, the phospholipid composition of OMVs may differ significantly from that of the OM itself. In summary, relying on the OMV composition as proxy for the OM composition may be misleading, and statements like the header in line 136 "The outer membrane lipid composition depends on Mpc" are not necessarily backed up by the experimental data shown in the manuscript. Ideally, the authors would compare the phospholipid compositions of the IM and OM.

Minor comments:

1. There are other instances where the conclusions are not fully supported by the experimental results. For example, in line 203: While the pull-down assay does support the existence of an McpEFDA complex, this specific experiment does not demonstrate "that McpEFDA form a stable complex, from the cytoplasm to the OM".

2. Fig. 3A: Letters and numbers on the X and Y axis are too small.

3. Fig. 3A and B: The colors used are difficult to distinguish, particularly in the printed version of the manuscript. Additionally, it is unfortunate that the order of the lipids changes between figures (CL, PG, ... in panel A, but PG, PE, PC, ... in panel B).

4. Fig. 3B: In case of the lysolipids, the mpc operon mutant and mpcA mutant exhibit effects opposite to those of the mpcF and mpcD mutants. How can this discrepancy be explained if the four proteins operate in the same system?

5. Line 114: What are "four more strongly genes"?

Lines 157 to 159: Check sentence.

Referee #2:

Report for the manuscript EMBOJ-2024-119476

The paper "A lipid transport Mla Pqi Chimeric system is essential for *Brucella abortus* survival in macrophages" report a very interesting story related to the identification and characterization of a potential system transporting glycerophospholipids between the inner and outer membranes of this bacterial pathogens. Most of the knowledge on the OM biogenesis and maintenance is coming from works on a very limited number of bacteria (and mostly in *E. coli*). There is therefore a strong interest in characterizing such systems in understudied bacteria.

The authors identified by a phenotypic screen of a TnSeq library of this pathogen a set of genes that are essential for *B. abortus* to survive in presence of sodium deoxycholate (DOC). Among the most interesting genes were four genes (in a potential operon they named mpc), with one of this gene encoding a protein with a MCE domain described to be involved in transport of glycerophospholipids. Markerless deletions (and complementation) of these genes confirmed the importance of these genes for growth with DOC. Additionally, the authors showed a slight increase of sensitivity to polymyxin B of these mutants. The authors assessed the survival of these mutants in macrophages and showed that while their intracellular growth was not affected, their survival was reduced at early time-points of infection, but not at later-time points. The authors determined the lipid composition of whole cells and potential outer membrane vesicles of these mutants and detected differences notably in the level of cardiolipin and of ornithine and lyso-ornithine lipids. A deletion of *olsA* in the Δmpc mutant restored the WT phenotype on DOC and in macrophages suggesting that the accumulation of lyso-ornithine could compensate the absence of the Mpc system. Interestingly, unlike what was observed in previous studies with mutations in AsmA-like proteins compensating for the default observed in Mla mutants, the authors showed no compensation by the *asmA* mutation in the Δmpc background. The authors then performed structural modelling of the MpcEFDA proteins and potential complex. While MpcEFD display homology to the MlaEFD complex, MpcA displays homology to the lipoprotein PqiC, hence the name of their system that they consider a hybrid system between the previously described Mla and Pqi systems. A complex between the MpcEFDA proteins is supported by pull-down experiment.

The paper is well written and pleasant to read. The results are very interesting.

Most of, but not all, the conclusions are supported by the results. It is necessary to provide more information on certain aspects of the study to fully support the author's conclusions, and some rewording might be necessary to tone down some of the not fully supported conclusions of the manuscript.

Major concerns:

1- In Figure 1B related to sensitivity to polymyxin B, the observed differences fall within the margin of error for this type of measurement. Would it be possible to show the results of the complemented strains? This would be much more convincing that the observed phenotype is true.

2- The authors provided interesting results on the lipid composition of OMVs but there is no formal proof in the manuscript that the preparation that they made corresponds to OMVs. Could the authors provide additional data that formally demonstrated the existence of the OMVs (for example, electronic microscopy or measure of particles size). This could also reveal that the deletion of mpc genes leads to a modification of OMVs production that is often seen when deleting genes involved in GPLs transport.

3- In Figure 3B and S4 there are apparent discrepancies in term of quantification of GPLs between the different mutants of the mpc genes. How can this be explained if the proteins belong to the same system? This should be discussed in the manuscript. How many biological replicates were used in these experiments?

4- In Fig. 4C some statistical analyses were apparently performed but there is no indication of statistical difference between the different samples including between WT and Δmpc mutant which is contradictory to what is shown in Fig. 2A. Therefore, it is difficult to interpret the results of these experiments. Is this an oversight on the part of the authors, or is there in fact no significant difference? If the latter is the case, the authors' conclusions should be reviewed or the experiments repeated in order to be able to conclude whether these results are really significant.

5- Concerning structural aspects of the study, the authors clearly indicated that there are models. The authors should be cautious on the raised conclusions. For example, line 272, it is not "the proposed structure of Mpc ..." but "the proposed structural model of Mpc ...". Same for line 274: "The most notable aspect of the structure is the interaction between the MpcA

lipoprotein and the hexameric coiled ..." should be "The most notable aspect of the predicted structure is the interaction between the MpcA lipoprotein and the hexameric coiled The presented complex in figure 5 corresponds to the model 2 proposed in Figure S7. In Figure S7 the authors provided the pLDDT map of each complex, but it is difficult with only this information to understand why the authors favor this model 2 over the other models. In practice, the overall confidence in predictions for multimers should be based on a combination of all the metrics that are available in AlphaFold, including both pTM and ipTM as well as pLDDT and PAE interaction maps. It is therefore recommended that the authors provided these information to justify their choice of model 2. The PAE maps can be annotated and colored using this site: <https://thecodingbiologist.com/tools/pae.html>. Some of the proposed models show a collapsing effect classically observed with AF3 on long sequences. One way of solving this problem is to try to model the complex in overlapping fragments and overlap them using the Super command in Pymol. Based on these predictions the authors conclude that the Mpc complex forms from the cytoplasm to the OM (for example see line 202-203). While this is likely, there is no formal (biochemical/localisation) proof that MpcA is attached to the outer membrane and that MpcDEF is an inner membrane complex. I do not request that such biochemical demonstration be carried out (because this is considerable amount and difficult work) but for the authors to be careful with the used terms.

6- Line 196-197 and 273-274: The "precise" length of the complex should be given compared to the estimated or calculated size of the periplasmic space in *Brucella* (in Fig 5 or S7/S8?). Line 205-206, the authors cannot say "The present study characterises a newly identified complex spanning the periplasmic space, namely the Mpc (Mla-Pqi chimeric) system, connecting IM to OM in *B. abortus*" but should say " The present study characterises a newly identified complex potentially spanning the periplasmic space, namely the Mpc (Mla-Pqi chimeric) system, connecting IM to OM in *B. abortus*".

7- Another aspect concerning model 2 is the indication of a MpcD hydrophobic closed tunnel that could allow the passage of GPLs. However, there is a strong debate in the field on whether such potential bridging complex/proteins encompass a closed tunnel or an open slide to accommodate the amphipathic nature of GPLs. Did all the models of the Mpc complex actually predict a closed tunnel or some rather displayed an open slide? This can be further discussed line 284-287.

8- Line 173-177, the authors pointed to the existence of this extra long alpha-helices that might help MpcD to cross the periplasm and to the conservation in Hyphomicrobiales like *Agrobacterium* or *Sinorhizobium*. The existence of such extended alpha-helices in MlaD protein (with or without a beta-barrel at the end) has been identified not only in several Hyphomicrobiales (Rhizobiales), but also in most of the diderm bacteria (see Fig 5 and Supp. Fig. S12/Fig. S14 in Grasekamp KP et al. PMID: 37993432; PMCID: PMC10665443). This is worth mentioning and can also be added in the discussion point related to the evolution of Mpc proteins.

9- The directionality of these Mla/AsmA systems is hotly debated. I think the authors should be more cautious about the possibility that AsmA-like proteins transport GPLs in a bidirectional way. Both in *E. coli* (Sutterlin et al 2016 PMID: 26929379; PMCID: PMC4801249 and Grimm et al 2020 PMID: 33046656; PMCID: PMC7604412) and in *V. parvula* (Grasekamp KP et al. PMID: 37993432; PMCID: PMC10665443), mutations in *asmaA*-like genes suppressed the OM sensitivity of *mia* mutants and ³²P-labeled GPLs accumulated in the IM of a triple $\Delta yhdP$ - $ydbH$ - $tamB$ *E. coli* mutant (Douglass et al 2022 PMID: 35226662; PMCID: PMC8912898) suggesting that AsmA-like proteins rather function as anterograde systems. The fact that an *asmaA* mutation does not suppress the phenotype of a *mpc* mutant is worth discussing as compared to different results obtained in *E. coli* and *V. parvula*.

10- Line 280-284: the discussion about the evolution of the Mpc system is very interesting and merits further discussion. It would be worth looking in which species MpcA/PqiC proteins are present together with long MpcD/MlaD (without beta-barrel) and to see how this co-distribution can add to the conclusion raised by Grasekamp et al (PMID: 37993432; PMCID: PMC10665443) on the ancestry of the MlaEFD back to the last bacterial common ancestor while PqiB/LetB is a recent invention. Could it be that indeed MpcA/PqiC is also ancestral and that in proteobacteria PqiAB/LetAB evolved from MlaDEF?

Minor concerns

11- I think an interesting discussion can be made on the name of the system. While the arguments provided by the authors to justify of a chimeric system seem globally appropriate, I think that extra information is needed to potentially fully support the decision to call this system Mpc and not Mla or Pqi. For example, did the authors check for the genetic context of this operon and the synteny of this system as compared for example to the one of the Mla and Pqi of *E. coli* or other proteobacteria? The authors indicated that the MpcA protein is homologous to the PqiC protein: first, they should provide maybe a comparison (side by side or overlapped) of monomeric and multimeric forms of these two proteins. What about the homology between MpcA and MlaA (this is not indicated in the manuscript)? Also, the authors should clarify what is the homology in sequence and/structure between MpcE and MlaE/PqiA (clarify what is said also for MlaB line 172= is it that MpcE is homologous to both MlaE and MlaB?); MpcF and MlaF ; MpcD and MlaD vs MpcD and PqiB/LetB. Some gene phylogenies for these different genes may support the authors hypothesis that the system is hybrid.

12- In the introduction, since one of the main interests of this manuscript is the study of OM biogenesis and maintenance in understudied bacteria, it seems that it would be fair to cite the work performed on these types of systems in bacteria other than *E. coli*: for Mla in other bacteria such as Burkholderia (Bernier SP et al. PMID: 29986943; PMCID: PMC6112004) ; in *Stenotrophomonas maltophilia* (Coves X et al. PMID: 38469346; PMCID: PMC10925693), *Bordetella pertussis* (de Jonge EF et al. PMID: 36518166; PMCID: PMC9743053) ; *Pseudomonas aeruginosa* (Zhou C et al. PMID: 33845086 and Kaur M et al. PMID: 39373473; PMCID: PMC11537012) ; in *Acinetobacter baumannii* (Powers MJ, Trent MS. PMID: 30087182; PMCID: PMC6130378 and Powers MJ et al. PMID: 32880370; PMCID: PMC7500953) ; and in *Veillonella parvula* (Grasekamp KP et al. PMID: 37993432; PMCID: PMC10665443).

13- the same is true for the role of AsmA-like proteins in non *E. coli* bacteria: role of TamB in phospholipid transport in other bacteria: in *P. aeruginosa* (Sposato D et al PMID: 38305166; PMCID: PMC10900882) and in *V. parvula* (Grasekamp KP et al. PMID: 37993432; PMCID: PMC10665443).

- 14- For the AsmA-like proteins the authors essentially cited the two reviews (Cooper et al 2023 and Kumar and Ruiz 2023). It would be fair to cite the original studies of Ruiz et al. 2021 (PMID: 34781743; PMCID: PMC8593681) and Douglass et al 2022 (PMID: 35226662; PMCID: PMC8912898).
- 15- The authors should make it clearer whether the MpcD protein is the sole MCE domain containing protein in *Brucella abortus*.
- 16- How were chosen the DOC and the polymyxin B concentrations to do the stress sensitivity assays?
- 17- In Figure 1A, there is a clear complementation of the phenotype for each independent mutants which seems to indicate no polar effects in the constructed mutants. This is good. Therefore, I think the description of the construction of the markerless mutants in the material and methods section deserves a bit more detail.
- 18- The authors did not discuss on why they think that the mpc deletion affects survival in macrophages at early time-points but not at later time-points.
- 19- In Figure 2A and in all other figures where it is possible the authors should show the independent replicates and not solely the average (or mean).
- 20- Does the MpcA-strep-tag protein is able to complement a $\Delta mpcA$ mutant? This will support the veracity of the results of the pull-down assays.
- 21- It could be interesting to show side by side comparison (or overlapped) of MpcA with PqiC (as monomer and multimer). This would also be interesting to have this comparison for the other Mpc proteins with Mla homologues, notably for MpcD with a long MlaD.
- 22- Line 232-240: the results/discussion related to AsmA-like proteins are interesting. How many AsmA-like proteins are present in *Brucella*? Is there a TamB-like protein genetically associated with a TamA/BamA protein? This is worth mentioning.
- 23- Line 289, maybe length of the channel is more appropriate than size of the channel.
- 24- Line 344-357: the complete list of identified genes and the sequencing data must be made available.
- 25- Line 469-521, pull-down assay. It is not clear what is the control in these experiments. The authors should provide the list of the other proteins that were likely associated with the Mpc complex if any.
- 26- In Fig.2A please show the individual replicates. In Fig. 2B how many replicates were used?
- 27- Y axis legends are missing in Fig. 3A and 3B.
- 28- Show the replicates in Fig. S5B.
- 29- Show the replicates in Fig. 4C.

Additional non-essential suggestions for improving the study.

- 30- It would be interesting to look whether other Hyphomicrobiales contains also only one MCE domain containing protein.
- 31- To further support the role of Mpc proteins in the OM biogenesis and envelope stress, it might also be interested to assess the effect of other antibiotics that are used in some other phospholipid transport studies such as vancomycin, erythromycin or novobiocin that are large or charged antibiotics.

Referee #3:

In this paper the authors characterize the mpcEFDA operon from *Brucella abortus*, which is an mce protein system. This is an interesting system homologous to systems from other double-membraned bacteria, though to be lipid transporters. The authors convincingly show that in *B. abortus*, the mce system results in outer membrane sensitivity, as expected. Infection in macrophages is used to assess the importance for bacterial survival in macrophages, lipidomics is used to characterize lipid composition in OMVs of WT and mutant strains, a pull down experiment shows interaction of the mce proteins as expected, and finally, AlphaFold predictions are presented. I thank the authors for making their work available as a preprint for the community. The paper is interesting overall. My main concerns are related to inconsistency of conclusions with data presented, and over-interpretation of AlphaFold models, without experimental validation. I have outlined these below, and hope that it will be helpful for the authors.

Specific comments

- 1) This system is most homologous to Mce1 from mycobacteria, the Mce system from *V. parvula* and the Mce system from chloroplasts (TGD). The main differences are that the mce protein itself is a hetero vs. homo oligomer, and that the lipoprotein (MpcA) is not conserved. As there are many variations on Mce systems, a thorough comparison of the systems in the discussion would be informative, especially with Mce1 from mycobacteria, and TGD-like systems. While only a limited comparison is possible structurally, since there is not a structure of this system, this would be suitable for the discussion section.
- 2) As TGD systems have already been characterized (eg. PMCID: PMC3375562), and the system described here is most similar to the TGD system from chloroplasts, I suggest naming this a TGD-like system. It would be greatly valuable in avoiding confusion in the field, with multiple names for similarly grouped systems.

Both comments 1 and 2 more broadly relate to requesting the authors to more clearly and accurately put this work in the context of the field, and clearly note similarities to/differences from Mce systems that already exist and have been well characterized.

- 3) Fig 1B - please expand the legend to accurately explain what the reader is looking at

- 4) Line 125/Fig 2 - "The MPC system is required for survival in macrophages". The title/conclusion of this subsection is not consistent with the data presented. The mutants have a very small effect on survival in macrophages, especially at 24 and 48 hours, where it seems negligible. Confusingly, deletion of the full operon has the least effect. I would urge the authors to reconsider their interpretation of the data.
- 5) Previous work has suggested a difference in OMV production in Mce mutants (PMID: 26806181). Since the authors are characterizing OMVs, was this also observed? How does any change in the quantity of OMVs impact the lipidomics analysis and what would the authors expect the effect to be if there is a difference in total OMV production?
- 6) I did not follow the logic of the following conclusion, it is possible that I missed something, and would appreciate clarification (lines 137-145): because mce operon mutants have less cardiolipin and similar levels of PLs in OMVs, this system may act as a PL transport system
- 7) Use of AlphaFold models, and conclusions drawn. while AF predictions have revolutionized our ability to understand biology, they are hypothesis generators, not pieces of data. As such, it is imperative that conclusions based on AF predictions are validated experimentally. As evidenced by Fig. S7, S8 and the discussion from the authors, many different AF predictions are generated for this particular complex. If the authors would like to make hard conclusions from these (such as bridging of the IM and OM, oleic-acid mediated conformations, hydrophobicity, etc.) then further experimental data are needed to validate predictions, test hypotheses, and arrive at conclusions. If, on the other hand, the authors choose not to perform followup experiments and validation, then the interpretation and conclusions need to be tempered, and largely moved to the discussion, as they are valuable but speculative. As an example, Line 168 "Mpc bridges inner to outer membrane" - there are no data speaking to this conclusion. The only thing presented here is an alphafold model - it would be reasonable to speculate that this is a possibility in the discussion. However, even an experimental structure outside the context of inner and outer membranes would not be sufficient to draw a conclusion with that level of certainty.
- 8) Fig 5 - some inconsistency between the legend and the figure. It is not clear what the break in the AlphaFold predictions are towards the lipoprotein. The coloring on the Mce protein of B is not explained, and the key is not explained, and the interior shown does not look hydrophobic, if the colors are supposed to be a hydrophobicity scale. Since this is an AlphaFold prediction, discussing hydrophobicity (which will depend on rotamer conformation) may or may not be accurate.
- 9) The V. Parvula system shown in Fig. 6 is much more similar to the current system than the Mla system, and both are TGD-like systems - it is unfortunate that the authors named it Mla, but for the readers of the paper, and the field, this is likely to be confusing. It also lacks an MlaC homolog, as mpcEFDA does.

Namur, April 4, 2025

Dear Editor,
Dear Reviewers,

Hereunder is our point-by-point response to reviewers comments. We appreciated the comments of the reviewers and their interest for our data. We recognize that we have gone too far in some of our interpretations, as you will see in the answers provided below. We also performed most of the experiments proposed by the reviewers. The comments of the reviewers are indicated in blue, and our answers in black.

Referee #1:

Despite its critical role in Gram-negative bacteria, the mechanism of phospholipid transport from the inner membrane to the outer membrane remains poorly understood, particularly in species other than *E. coli*. Lannoy et al. identified a potential lipid transport system in *Brucella abortus* using a Tn-Seq approach to screen for detergent-resistant mutants. This so-called Mpc complex appears to be a hybrid of the known Mla and Ppi systems, presumably forming a bridge between the IM and OM. Computational predictions suggest that the Mpc system is a streamlined system comprising only four distinct subunits, and pull-down experiments suggest interactions among these proteins.

This is an interesting and relevant study that has the potential to represent a significant advance in the field. The manuscript is well written; however, my concern is that the conclusions regarding the OM composition are not sufficiently supported by the data (as outlined below). Furthermore, a substantial portion of the proposed model depends heavily on computational predictions.

Major concern:

The authors demonstrate that the absence of the Mpc system alters the lipid composition of OMVs. I was wondering whether OMVs accurately reflect the composition of the OM. My understanding is that OMVs can originate from vastly diverse processes such as blebbing or explosive lysis, and one of their functions is expel "waste" components (LPS, proteins, ...) from the OM. Moreover, since lipids are unevenly distributed within the membrane, the phospholipid composition of OMVs may differ significantly from that of the OM itself. In summary, relying on the OMV composition as proxy for the OM composition may be misleading, and statements like the header in line 136 "The outer membrane lipid composition depends on Mpc" are not necessarily backed up by the experimental data shown in the manuscript.

Ideally, the authors would compare the phospholipid compositions of the IM and OM.

We concur with the reviewer, we must be more cautious when using OMVs composition as a proxy for OM lipid composition. We therefore modified the title initially at line 136, which is now "The OMVs lipid composition depends on Mpc" (line 153). The correlation between lipid composition of the OM and the OMVs is now proposed in the discussion (lines 269-271). We do not think that OMVs are mainly generated by explosive lysis, mixing OM and IM fragments, since we found differences between whole cells extracts and OMVs regarding lipid composition. We did not try to separate IM from OM with classical protocols because this type of extraction is notoriously unsuccessful (Afzal *et al.* 1987 *J. Clin. Microbiol.* **25**, 2132). This is why we tried a new protocol (see below). Since the molecular biology of *B. abortus* is under-investigated, we did not have access to an antibody to detect IM proteins. For the purpose of this manuscript revision, we therefore engineered a *B. abortus* strain in which the FtsH protease was genetically fused to a 3Flag tag. FtsH is an IM protein with two

transmembrane domains, like the well described FtsH from *Escherichia coli*. The left panel of the figure below displays a Western blot against the Flag epitope (FtsH-3Flag is expected at 73.5kDa) and Omp25 (22.2kDa), an integral OM protein, on lysates from the wild type (WT) and 4 independently engineered strains (#1 to #4). Omp25 is detected in all strains, while the FtsH-3flag is only detected in the engineered strains, as expected. The right panel presents a Western blot against the FtsH-3Flag and Omp25, on different fractions of the membrane separation assay, according to a recently published method (Giovannercole *et al. J Mol Biol.* 2025 PMID: 40043834). In the soluble fraction, cytoplasmic and periplasmic proteins (cyto-peri) were found to exhibit only a faint band, while in other fractions, i.e. inner and outer membranes (IOM), inner membrane (IM) and outer membrane (OM), both proteins were detected in fusion strains across all membrane fractions. This data indicates that the membrane separation assay was not efficient. Our conclusion is thus that currently there is no efficient protocol available to separate IM from OM, and thus we adapted our manuscript to avoid the confusion between OM and OMs.

Minor comments:

1. There are other instances where the conclusions are not fully supported by the experimental results. For example, in line 203: While the pull-down assay does support the existence of an McpEFDA complex, this specific experiment does not demonstrate "that McpEFDA form a stable complex, from the cytoplasm to the OM".

We agree with the reviewer. Our pull-down data show strong interactions between MpcA, MpcD, MpcE and MpcF, but they do not prove that MpcA is attached to the OM while it is reasonable to assume that the ATPase is in the cytoplasm and that MpcE is an integral IM protein. In the revised manuscript, the formation of a stable complex bridging the two membranes is a model proposed as an interpretation of the experimental data, which are also supported by the homology analysis, MpcA being a lipoprotein similar to PqiC, which is recognized as an OM lipoprotein (Cooper *et al. 2024 EMBO Rep.* **25**, 82). Moreover, the alpha-helices tunnel formed by hexameric MpcD is similar to the structure proposed for hexameric PqiB, which inserts into the PqiC octameric ring. Therefore, we now write that our prediction is that the Mpc could bridge IM to OM (lines 24 and 188) and we developed our

conclusions more carefully at the end of Results section (lines 232-237 in the revised manuscript).

2. Fig. 3A: Letters and numbers on the X and Y axis are too small.

This is corrected in the revised manuscript, we changed Figure 3 according to another comment below, it is now Figure 4.

3. Fig. 3A and B: The colors used are difficult to distinguish, particularly in the printed version of the manuscript. Additionally, it is unfortunate that the order of the lipids changes between figures (CL, PG, ... in panel A, but PG, PE, PC, ... in panel B).

Figure 3B has been completely reformatted to answer to a comment below, it is now Figure 4B. The colors are adapted in Figure 4A, in the revised manuscript.

4. Fig. 3B: In case of the lysolipids, the *mpc* operon mutant and *mpcA* mutant exhibit effects opposite to those of the *mpcF* and *mpcD* mutants. How can this discrepancy be explained if the four proteins operate in the same system?

We agree. That does not make sense if the hypothesis is that Mpc is transporting a specific set of lipids. Actually, we realized that the absence of Mpc generates subtle changes in lipids composition, which are better illustrated in the revised manuscript (Figure 4B). We think that the change in lipid composition, happening in the IM, in the OM or in both membranes, could generate pleiotropic effects. It is also clear that all mutants are not equal. For example, the $\Delta mpcD$ and Δmpc_operon mutants are less affected for growth on rich medium (without DOC) compared to the other mutants (see Figure 2A), suggesting that the presence of MpcD in the absence of another component could be toxic. It is therefore likely that each mutant adapted its envelope to face different indirect effects of the mutations, leading to perturbations in lipids composition. However, the decreased abundance of CL in the OMVs of the mutants, as well as OL in whole cells, is still found in all mutants and statistically significant, together with the decreased abundance of LOL in both OMVs and whole cells. In the revised manuscript (lines 166-172), we thus limit our interpretation to these variations.

5. Line 114: What are "four more strongly genes"?

It was changed to "the four more strongly attenuated genes" in the revised manuscript (line 126)

Lines 157 to 159: Check sentence.

We replaced "in the IM. The phenotype" by "in the IM, the phenotype" in the revised manuscript (line 174)

Referee #2:

Report for the manuscript EMBOJ-2024-119476

The paper "A lipid transport Mla Pqi Chimeric system is essential for *Brucella abortus* survival in macrophages" report a very interesting story related to the identification and characterization of a potential system transporting glycerophospholipids between the inner and outer membranes of this bacterial pathogens. Most of the knowledge on the OM biogenesis and maintenance is coming from works on a very limited number of bacteria (and mostly in *E. coli*). There is therefore a strong interest in characterizing such systems in understudied bacteria.

The authors identified by a phenotypic screen of a TnSeq library of this pathogen a set of genes that are essential for *B. abortus* to survive in presence of sodium deoxycholate (DOC). Among the most interesting genes were four genes (in a potential operon they named *mpc*), with one of this gene encoding a protein with a MCE domain described to be involved in transport of glycerophospholipids. Markerless deletions (and complementation) of these genes confirmed the importance of these genes for growth with DOC. Additionally, the authors showed a slight increase of sensitivity to polymyxin B of these mutants. The authors assessed the survival of these mutants in macrophages and showed that while their intracellular growth was not affected, their survival was reduced at early time-points of infection, but not at later-time points. The authors determined the lipid composition of whole cells and potential outer membrane vesicles of these mutants and detected differences notably in the level of cardiolipin and of ornithine and lyso-ornithine lipids. A deletion of *olsA* in the Δmpc mutant restored the WT phenotype on DOC and in macrophages suggesting that the accumulation of lyso-ornithine could compensate the absence of the Mpc system. Interestingly, unlike what was observed in previous studies with mutations in AsmA-like proteins compensating for the defect observed in Mla mutants, the authors showed no compensation by the *asmA* mutation in the Δmpc background. The authors then performed structural modelling of the MpcEFDA proteins and potential complex. While MpcEFD display homology to the MlaEFD complex, MpcA displays homology to the lipoprotein PqiC, hence the name of their system that they consider a hybrid system between the previously described Mla and Pqi systems. A complex between the MpcEFDA proteins is supported by pull-down experiment.

The paper is well written and pleasant to read. The results are very interesting. Most of, but not all, the conclusions are supported by the results. It is necessary to provide more information on certain aspects of the study to fully support the author's conclusions, and some rewording might be necessary to tone down some of the not fully supported conclusions of the manuscript.

We thank the reviewer for his/her detailed analysis of the manuscript and the positive comments about the work.

Major concerns:

In Figure 1B related to sensitivity to polymyxin B, the observed differences fall within the margin of error for this type of measurement. Would it be possible to show the results of the complemented strains? This would be much more convincing that the observed phenotype is true.

For the revised version of this manuscript, tests for polymyxin B sensitivity were conducted in the presence of complemented strains. However, the replicated tests did not confirm the results of the initial tests, and the complemented strains did not complement the phenotype. Furthermore, the replicates were inconsistent. In view of these data, it was decided to remove the polymyxin B assay from the manuscript.

2- The authors provided interesting results on the lipid composition of OMVs but there is no formal proof in the manuscript that the preparation that they made corresponds to OMVs. Could the authors provide additional data that formally demonstrated the existence of the OMVs (for example, electronic microscopy or measure of particles size). This could also reveal that the deletion of *mpc* genes leads to a modification of OMVs production that is often seen when deleting genes involved in GPLs transport.

Our data do not indicate a drastic change in the amount of OMVs, as seen in Grasekamp et al (2023). However, we detected small changes in the proportion of OMVs (in dry weight), as reported in this Table :

Percentage of pellet (corresponding to the whole cells extracts) and outer membrane vesicles (OMVs) were obtained during the isolation process. During the isolation of OMVs, the pellet and the isolated OMVs were lyophilised and weighed. This table presents the weight percentage of pellet and OMVs from the 4.8 L of 48 hours cultures.

	Ratio weight dry pellet (%)	Ratio weight dry OMVs (%)
WT	97.67	2.33
$\Delta mpcE$	93.53	6.47
$\Delta mpcF$	93.73	6.27
$\Delta mpcD$	96.23	3.72
$\Delta mpcA$	95.56	4.44
Δmpc operon	96.87	3.13
Omp25-2b D2A	94.18	5.82
WT DOC 0.01%	97.27	2.73

Since the differences are rather small, we decided to not include these values in the revised manuscript.

As suggested by this reviewer, we also tried scanning electron microscopy. However, the image that we obtained is not very informative (see below), and thus we decided that we would not include it in the revised manuscript.

We think that our preparations of “OMVs” could also contain OM fragments but are probably not a mix of IM and OM generated by lysis, since we have distinct lipid compositions between whole cells and “OMVs” shown in Figure 4B. Each OMV preparation was made from 4.8 liters of culture in a BSL3 facility in Pamplona. It is thus a risky experiment (there are more than 3×10^9 bacteria per milliliter and the infectious dose is around 10^3 - 10^4) that we choose to not repeat.

Since we also worry about the nature of the OMVs, we undertook a proteomic analysis of the OMVs preparations. When we compared the proteomic composition of the OMVs with the whole cells extracts, it was similar. Since we previously showed that many beta-barrels in the OM are covalently linked to peptidoglycan (Godessart *et al.* 2021 *Nat. Microbiol.* **6**, 27), it is actually not surprising that OMVs contain very few OM proteins, probably hidden in the background. Therefore, using the same OMVs preparation protocol, we analyzed the proteomic content of OMVs generated by a strain in which the linkage to the peptidoglycan has been mutated in two major OM proteins (Omp25 and Omp2b, the *omp25*_{D2A}*omp2b*_{D2A} in Godessart *et al.*, 2021). In this case, the OMV fraction is enriched for Omp25 and Omp2b (2.6- and 3.5-fold, respectively), suggesting that the OMV protocol indeed allow the generation of OMV and/or OM fragments.

3- In Figure 3B (now Fig. 4B) and S4 (now appendix fig. S3) there are apparent discrepancies in term of quantification of GPLs between the different mutants of the *mpc* genes. How can this be explained if the proteins belong to the same system? This should be discussed in the manuscript. How many biological replicates were used in these experiments?

We thank the reviewer for this comment. Indeed, we realized that the representation of the data was misleading and for some data, inconsistent. This inconsistency is however

exacerbated by the way we presented the data in the original manuscript. In the revised manuscript the absolute concentrations are presented, and they suggest that inconsistency was limited to a low number of lipids. The inconsistencies were also pointed by reviewer 1, and as indicated in the answer to comment 4 of reviewer #1, our data suggest that all mutants are not equal, and some have probably adapted their lipid composition in different ways. As indicated above, these experiments involve huge volumes of culture for such a class III pathogen, therefore we considered the 5 different mutants of the system as biological replicates of the absence of the Mpc complex, and we used the wild type strain as the reference in the statistical analysis (a compliance test). We also limited our interpretations to the cases in which all 5 mutants are affected similarly compared to the wild type strain. Finally, it could be mentioned that the lower amount of lyso-ornithine-lipid (LOL) in the mutants can be correlated to the higher susceptibility to DOC, and since a strain accumulating LOL is protected from DOC like the wild type strain, it also argues in favor of a true decrease of LOL in the mutants compared to the wild type. The accumulation of LOL detected by TLC in the $\Delta olsA$ mutant is now shown in the revised manuscript (Fig. 5A).

4- In Fig. 4C (now Fig. 5C) some statistical analyses were apparently performed but there is no indication of statistical difference between the different samples including between WT and Δmpc mutant which is contradictory to what is shown in Fig. 2A. Therefore, it is difficult to interpret the results of these experiments. Is this an oversight on the part of the authors, or is there in fact no significant difference? If the latter is the case, the authors' conclusions should be reviewed or the experiments repeated in order to be able to conclude whether these results are really significant.

It is our fault; we forgot to add the statistical analysis on this figure. It is indicated in the figure 5C of the revised manuscript. We apologize for this error in the original manuscript.

5- Concerning structural aspects of the study, the authors clearly indicated that there are models. The authors should be cautious on the raised conclusions. For example, line 272, it is not "the proposed structure of Mpc ..." but "the proposed structural model of Mpc ...".

This is corrected in the revised manuscript (line 314).

Same for line 274: "The most notable aspect of the structure is the interaction between the MpcA lipoprotein and the hexameric coiled ..." should be "The most notable aspect of the predicted structure is the interaction between the MpcA lipoprotein and the hexameric coiled

This is corrected in the revised manuscript (line 317).

The presented complex in figure 5 corresponds to the model 2 proposed in Figure S7. In Figure S7 the authors provided the pLDDT map of each complex, but it is difficult with only this information to understand why the authors favor this model 2 over the other models.

In terms of pLDDT map, model 2 is very close to other models, but model 2 is very similar to the structures revealed by the study of Cooper *et al.* (2024 *EMBO Rep.* **25**, 82), where the PqiC octamer (homologous to MpcA) is interacting with the C-terminal part of the PqiB hexamer (homologous to MpcD, except that MpcD has only one MCE domain while PqiB has three). Thus model 2 is supported by AlphaFold (AF) modeling **and** homology analysis, it is not an AF-only prediction.

In practice, the overall confidence in predictions for multimers should be based on a combination of all the metrics that are available in Alphafold, including both pTM and ipTM as well as pLDDT and PAE interaction maps. It is therefore recommended that the

authors provided these information to justify their choice of model 2. The PAE maps can be annotated and colored using this site: <https://thecodingbiologist.com/tools/pae.html>.

This is now reported in the revised manuscript (Fig. EV4 and Appendix Fig. S4)

Some of the proposed models show a collapsing effect classically observed with AF3 on long sequences. One way of solving this problem is to try to model the complex in overlapping fragments and overlap them using the Super command in Pymol.

We applied the recommended procedure, and it is now shown in the revised manuscript. We thank the reviewer for her/his recommendations.

Based on these predictions the authors conclude that the Mpc complex forms from the cytoplasm to the OM (for example see line 202-203). While this is likely, there is no formal (biochemical/localisation) proof that MpcA is attached to the outer membrane and that MpcDEF is an inner membrane complex. I do not request that such biochemical demonstration be carried out (because this is considerable amount and difficult work) but for the authors to be careful with the used terms.

We agree with this comment, the manuscript has been checked (lines 24, 188 and 232-237), the bridge of the IM to the OM is now presented as the mostly likely interpretation, and we are less affirmative in the revised manuscript about this aspect.

6- Line 196-197 and 273-274: The "precise" length of the complex should be given compared to the estimated or calculated size of the periplasmic space in *Brucella* (in Fig 5 or S7/S8?).

The size of the periplasm was estimated by Godessart *et al.* 2021 (Extended data Fig. 5b in this published paper), it is comprised between 25 and 35 nm according to Cryo-EM images, which is consistent with periplasm size proposed in other bacteria. This is now indicated in the revised manuscript (lines 222-224).

Line 205-206, the authors cannot say "The present study characterises a newly identified complex spanning the periplasmic space, namely the Mpc (Mla-Pqi chimeric) system, connecting IM to OM in *B. abortus*" but should say "The present study characterises a newly identified complex potentially spanning the periplasmic space, namely the Mpc (Mla-Pqi chimeric) system, connecting IM to OM in *B. abortus*".

This is corrected in the revised manuscript (line 239).

7- Another aspect concerning model 2 is the indication of a MpcD hydrophobic closed tunnel that could allow the passage of GPLs. However, there is a strong debate in the field on whether such potential bridging complex/proteins encompass a closed tunnel or an open slide to accommodate the amphipathic nature of GPLs. Did all the models of the Mpc complex actually predict a closed tunnel or some rather displayed an open slide? This can be further discussed line 284-287.

We did not detect an open slide in any of the models, but it does not mean that it is not possible. We know nothing about the dynamics of these structures, and it may well be that predicted structures correspond to an inactive state. Intriguingly, when the oleic acid was included in the model, the charged carboxylate groups are oriented towards the junctions between the helices. We thus remain careful in the Discussion about this aspect, but it is now mentioned in the revised manuscript (lines 332-335).

8- Line 173-177, the authors pointed to the existence of this extra long alpha-helices that might help MpcD to cross the periplasm and to the conservation in Hyphomicrobiales like *Agrobacterium* or *Sinorhizobium*. The existence of such extended alpha-helices in MlaD protein (with or without a beta-barrel at the end) has been identified not only in several Hyphomicrobiales (Rhizobiales), but also in most of the diderm bacteria (see Fig 5 and Supp. Fig. S12/Fig. S14 in Grasekamp KP et al. PMID: 37993432; PMCID: PMC10665443). This is worth mentioning and can also be added in the discussion point related to the evolution of Mpc proteins.

Indeed, this is now added in the revised manuscript (lines 324-330). We also provide a synteny analysis of the *mpc* operon in alpha-proteobacteria (Appendix Figure S5), which suggests that extended alpha-helices without beta-barrel are frequently associated with MpcA/PqiC-like lipoproteins.

9- The directionality of these Mla/AsmA systems is hotly debated. I think the authors should be more cautious about the possibility that AsmA-like proteins transport GPLs in a bidirectional way. Both in *E. coli* (Sutterlin et al 2016 PMID: 26929379; PMCID: PMC4801249 and Grimm et al 2020 PMID: 33046656; PMCID: PMC7604412) and in *V. parvula* (Grasekamp KP et al. PMID: 37993432; PMCID: PMC10665443), mutations in *asmA*-like genes suppressed the OM sensitivity of *mia* mutants and 32P-labeled GPLs accumulated in the IM of a triple $\Delta yhdP$ - $ydbH$ - $tamB$ *E. coli* mutant (Douglass et al 2022 PMID: 35226662; PMCID: PMC8912898) suggesting that AsmA-like proteins rather function as anterograde systems. The fact that an *asmA* mutation does not suppress the phenotype of a *mpc* mutant is worth discussing as compared to different results obtained in *E. coli* and *V. parvula*.

It seems that our description of the “*asmA* case” has been oversimplified in our original manuscript. We apologize for this. There are actually at least three AsmA homologs predicted from the *B. abortus* genome, TamB, a long AsmA and a short AsmA. The only “*asmA*” identified in our Tn-seq on the wild type strain on DOC is potentially coding for the short AsmA. This gene was also identified in other Tn-seq screens, in mice (Barbieux *et al.* 2024 *PLoS Pathog.* PMID: 39186777). The double mutant $\Delta mpc_operon\Delta asmA$ gives a very slight suppression compared to the Δmpc_operon strain (Fig. EV2 of the revised manuscript), and we removed the data obtained with the $\Delta mpc_operon\Delta asmA$ is infection because dilutions of the CFU counting were not reliable (we think that maybe this strain adheres to the plastic of the tips used for pipetting). Overall, we think that our data do not clearly support the absence of suppression of Δmpc_operon with an *asmA* mutation, only the Tn-seq performed on the Δmpc_operon strain suggests this absence of suppression. We thus decided to not incorporate this discussion in the revised manuscript.

10- Line 280-284: the discussion about the evolution of the Mpc system is very interesting and merits further discussion. It would be worth looking in which species MpcA/PqiC proteins are present together with long MpcD/MlaD (without beta-barrel) and to see how this co-distribution can add to the conclusion raised by Grasekamp et al (PMID: 37993432; PMCID: PMC10665443) on the ancestry of the MlaEFD back to the last bacterial common ancestor while PqiB/LetB is a recent invention. Could it be that indeed MpcA/PqiC is also ancestral and that in proteobacteria PqiAB/LetAB evolved from MlaDEF?

The results now available in the revised manuscript (Appendix Fig. S5) show that the *mpc* operon is syntenic with loci in several Hyphomicrobiales as well as other alpha-proteobacteria like *Rhodobacter sphaeroides*. Thus, it seems that the Mpc organization is ancestral, and these data are in agreement with the proposition of Grasekamp et al. This is now indicated in the revised manuscript (lines 324-330 and 337-339).

Minor concerns

11- I think an interesting discussion can be made on the name of the system. While the arguments provided by the authors to justify of a chimeric system seem globally appropriate, I think that extra information is needed to potentially fully support the decision to call this system Mpc and not Mla or Pqi. For example, did the authors check for the genetic context of this operon and the synteny of this system as compared for example to the one of the Mla and Pqi of *E. coli* or other proteobacteria?

As indicated in the previous response, we indeed found synteny with alpha-proteobacteria. As requested, we analyzed the genetic context outside of the *mpc* operon and we realized that it is not informative. The Appendix Fig. S5 of the revised manuscript clearly shows that Mpc is a chimeric system composed of Mla and Pqi homologs. We thus think that this name is appropriate, given the previous knowledge about the structure of Mla and Pqi systems.

The authors indicated that the MpcA protein is homologous to the PqiC protein: first, they should provide maybe a comparison (side by side or overlapped) of monomeric and multimeric forms of these two proteins.

We thank the reviewer for this suggestion. The superimposition of MpcA (predicted) and PqiC structures are now shown in Appendix Fig. S6AB. For this analysis, the structural similarity is very clear, RMSD values are $<2\text{\AA}$ after superimposition.

What about the homology between MpcA and MlaA (this is not indicated in the manuscript)?

We did not find homology between MpcA and MlaA, it is now clearly indicated in the revised manuscript (lines 203-204) and Appendix Figure S5A

Also, the authors should clarify what is the homology in sequence and/structure between MpcE and MlaE/PqiA (clarify what is said also for MlaB line 172= is it that MpcE is homologous to both MlaE and MlaB?); MpcF and MlaF ; MpcD and MlaD vs MpcD and PqiB/LetB. Some gene phylogenies for these different genes may support the authors hypothesis that the system is hybrid

We did not realize that it was so unclear, and we agree with the reviewer that it is important to show that Mpc is indeed a chimeric system. We now provide a comparison of the different components of the Mpc system with members of the Mla and PqiC, in Appendix Figure S5. This is clarifying the chimeric/hybrid nature of the Mpc system compared to Mla and Pqi.

12- In the introduction, since one of the main interests of this manuscript is the study of OM biogenesis and maintenance in understudied bacteria, it seems that it would be fair to cite the work performed on these types of systems in bacteria other than *E. coli*: for Mla in other bacteria such as Burkholderia (Bernier SP et al. PMID: 29986943; PMCID: PMC6112004) ; in Stenotrophomonas maltophilia (Coves X et al. PMID: 38469346; PMCID: PMC10925693), Bordetella pertussis (de Jonge EF et al. PMID: 36518166; PMCID: PMC9743053) ; Pseudomonas aeruginosa (Zhou C et al. PMID: 33845086 and Kaur M et al. PMID: 39373473; PMCID: PMC11537012) ; in Acinetobacter baumannii (Powers MJ, Trent MS. PMID: 30087182; PMCID: PMC6130378 and Powers MJ et al. PMID: 32880370; PMCID: PMC7500953) ; and in Veillonella parvula (Grasekamp KP et al. PMID: 37993432; PMCID: PMC10665443).

We warmly thank the reviewer for these suggestions, these references have been all included in the revised manuscript (lines 65-70).

13- the same is true for the role of AsmA-like proteins in non *E. coli* bacteria: role of TamB in phospholipid transport in other bacteria: in *P. aeruginosa* (Sposato D et al PMID: 38305166; PMCID: PMC10900882) and in *V. parvula* (Grasekamp KP et al. PMID: 37993432; PMCID: PMC10665443).

These references are now included in the revised manuscript (lines 45-48).

14- For the AsmA-like proteins the authors essentially cited the two reviews (Cooper et al 2023 and Kumar and Ruiz 2023). It would be fair to cite the original studies of Ruiz et al. 2021 (PMID: 34781743; PMCID: PMC8593681) and Douglass et al 2022 (PMID: 35226662; PMCID: PMC8912898).

This is true, original papers are now cited in the revised manuscript (lines 45-48).

15- The authors should make it clearer whether the MpcD protein is the sole MCE domain containing protein in *Brucella abortus*.

MpcD is the only predicted protein with a MCE domain, that we were able to detect from the *B. abortus* genome analysis. This now indicated in the revised manuscript (lines 102-103).

16- How were chosen the DOC and the polymyxin B concentrations to do the stress sensitivity assays?

We tested several concentrations before undertaking the Tn-seq. We chose the highest concentration of DOC at which the wild type strain was not impaired for growth on rich medium (see the figure below). This is indicated in the Materials and Methods section of the revised manuscript (lines 379-380). The effect of polymyxin B is tested with E-test strips with a range of concentrations, thus it was not needed to adjust the polymyxin B concentration.

17- In Figure 1A, there is a clear complementation of the phenotype for each independent mutants which seems to indicate no polar effects in the constructed mutants. This is good. Therefore, I think the description of the construction of the markerless mutants in the material and methods section deserves a bit more detail.

The construction of markerless mutants is described in the Materials and Methods of the revised manuscript, and the absence of an antibiotic resistance cassette is indicated at lines 359-361.

18- The authors did not discuss on why they think that the *mpc* deletion affects survival in macrophages at early time-points but not at later time-points.

We can be only very speculative at this stage. We hypothesize that a stress occurs in endosomal compartments, shortly after entry, that is bactericidal for the *mpc* mutants. It is

likely that this stress alters the envelope of the bacterium, but the nature of this stress is unknown. It may be due to antimicrobial peptides, but we did not have the time to develop this aspect in the current study. This is briefly mentioned in the revised manuscript (line 251).

19- In Figure 2A and in all other figures where it is possible the authors should show the independent replicates and not solely the average (or mean).

The revised figures take this suggestion into account.

20- Does the MpcA-strep-tag protein is able to complement a $\Delta mpcA$ mutant? This will support the veracity of the results of the pull-down assays.

We tested the function of two fusions of MpcA with the Strep tag (strep-MpcA and MpcA-strep). The MpcA-strep fusion was the only one to be functional. We replaced the *mpcA* gene by the *mpcA-strep* fusion and this strain displays the same sensitivity to DOC as the wild type (while the fusion strep-MpcA generated a phenotype similar to the Δmpc mutants). This is why we used the *mpcA-strep* fusion for the pulldown analysis. This is now indicated in the revised manuscript (lines 229-230)

21- It could be interesting to show side by side comparison (or overlapped) of MpcA with PqiC (as monomer and multimer). This would also be interesting to have this comparison for the other Mpc proteins with Mla homologues, notably for MpcD with a long MlaD.

Both are reported in Appendix Figure S6 of the revised manuscript.

22- Line 232-240: the results/discussion related to AsmA-like proteins are interesting. How many AsmA-like proteins are present in Brucella? Is there a TamB-like protein genetically associated with a TamA/BamA protein? This is worth mentioning.

As indicated above, there are at least three AsmA homologs predicted from the *B. abortus* genome, TamB, a long AsmA and a short AsmA. TamB is encoded from the same operon as a TamA homolog, and Tn-seq data suggest that *tamAB* could play a role in macrophages infection (Sternon *et al.*, 2018 *Infect. Immun.* PMID: 29844240). This is now reported in the revised manuscript (lines 272-278).

23- Line 289, maybe length of the channel is more appropriate than size of the channel.

This is corrected in the revised manuscript (lines 314-315).

24- Line 344-357: the complete list of identified genes and the sequencing data must be made available.

This is now provided with the revised manuscript (Appendix Table S5.). All the data are also presented in the form of a graph at Fig. 1.

25- Line 469-521, pull-down assay. It is not clear what is the control in these experiments. The authors should provide the list of the other proteins that were likely associated with the Mpc complex if any.

The control was the wild type strain, i.e. without a strep tag fused to MpcA. Therefore, all contaminants should be shared with this control, and MpcA-strep interactors should be identified more often in the extracts of the strain producing MpcA-strep compared to the wild type control. Besides MpcA, we found only MpcD, MpcE and MpcF. It does not mean that the Mpc system does not interact with anything else, but if it does, it is likely to be a less stable interaction.

26- In Fig.2A please show the individual replicates. In Fig. 2B how many replicates were used? Individual replicates are now shown in Fig. 3A of the revised manuscript. There were 3 biological replicates in Fig. 3B, each with 36 macrophages counted (thus a total of 108 macrophages). Counting was performed by two individuals. The very similar distribution for all the mutants is consistent with the absence of effect of the *mpc* mutation on the adhesion and entry of *B. abortus* in J774.A1 macrophages.

27- Y axis legends are missing in Fig. 3A and 3B.

We apologize for this mistake. A legend was added to Fig. 4A (Fig. 3A in the original manuscript). The old Fig. 3B is replaced by Fig. 4B, to better represent the data, as indicated in the response to comment 4 of reviewer #1.

28- Show the replicates in Fig. S5B.

Individual values are now shown in Fig. EV2B (Fig. S5B in the original manuscript).

29- Show the replicates in Fig. 4C.

Replicates are shown in Fig. 5C in the revised manuscript (replacing Fig. 4C of the original manuscript).

Additional non-essential suggestions for improving the study.

30- It would be interesting to look whether other Hyphomicrobiales contains also only one MCE domain containing protein.

We did not make a complete survey in Hyphomicrobiales, but analysis of homologs using MpcD (encoded BAB1_1040 in *B. abortus* 2308) with the MaGe database and Delta-Blast suggest that *Sinorhizobium meliloti* 1021 does have only one MCE-containing protein. Thus it seems that having a single MCE-containing homolog is not a feature linked to a reduced genome (Appendix Figure S5C).

31- To further support the role of Mpc proteins in the OM biogenesis and envelope stress, it might also be interested to assess the effect of other antibiotics that are used in some other phospholipid transport studies such as vancomycin, erythromycin or novobiocin that are large or charged antibiotics.

We started these experiments indeed, but until now the variability between antibiotics does not allow us to reach strong conclusions regarding envelope integrity. Therefore we decided to not include these data in the revised manuscript. However, we mention the growth in the presence of vancomycin at lines 133-137.

Referee #3:

In this paper the authors characterize the *mpeEFDA* operon from *Brucella abortus*, which is an *mce* protein system. This is an interesting system homologous to systems from other double-membraned bacteria, though to be lipid transporters. The authors convincingly show that in *B. abortus*, the *mce* system results in outer membrane sensitivity, as expected. Infection in macrophages is used to assess the importance for bacterial survival in macrophages, lipidomics is used to characterize lipid composition in OMVs of WT and mutant strains, a pull down experiment shows interaction of the *mce* proteins as expected, and finally, AlphaFold predictions are presented. I thank the authors for making their work available as a preprint for the community. The paper is interesting overall. My main concerns are related to inconsistency of conclusions with data presented, and over-interpretation of AlphaFold models, without experimental validation. I have outlined these below, and hope that it will be helpful for the authors.

Specific comments

1) This system is most homologous to *Mce1* from mycobacteria, the *Mce* system from *V. parvula* and the *Mce* system from chloroplasts (TGD). The main differences are that the *mce* protein itself is a hetero vs. homo oligomer, and that the lipoprotein (*MpcA*) is not conserved. As there are many variations on *Mce* systems, a thorough comparison of the systems in the discussion would be informative, especially with *Mce1* from mycobacteria, and TGD-like systems. While only a limited comparison is possible structurally, since there is not a structure of this system, this would be suitable for the discussion section.

There is major difference between the *Mce1* protein and the *Mpc* system presented here : while the *MpcA* lipoprotein is homologous to *PqiC*, the connection with the mycobacterial “outer membrane” of *Mycobacterium smegmatis* is made by the C-terminal domain of *Mce1*, the homolog of *MpcD*. In *B. abortus* and *Hyphomicrobiales* that we could analyze, *MpcD* is lacking this C-terminal domain. The shared homologies of *Mpc* for four *Mla* subunits and with *PqiC* for the *MpcA* lipoprotein, now reported in Appendix Figure S5-S6, support our proposition to name our system *Mla-Pqi*-chimeric. We apologize if it was not clear enough in our original manuscript. Reference to the *Mce1* and TGD2 is now made in the revised manuscript (lines 82-86).

2) As TGD systems have already been characterized (eg. PMID: PMC3375562), and the system described here is most similar to the TGD system from chloroplasts, I suggest naming this a TGD-like system. It would be greatly valuable in avoiding confusion in the field, with multiple names for similarly grouped systems

We are sorry that there are already different names for similar systems. As explained in the previous comments and responses to reviewer #2, our system is best characterized by its hybrid nature between the two previously characterized systems *Mla* and *Pqi*. We hope that our sequence comparisons, reported in Appendix Figure S5 of the revised manuscript, could convince this reviewer about the chimeric nature of the *Mpc* system.

Both comments 1 and 2 more broadly relate to requesting the authors to more clearly and accurately put this work in the context of the field, and clearly note similarities to/differences from *Mce* systems that already exist and have been well characterized.

3) Fig 1B - please expand the legend to accurately explain what the reader is looking at. The legend is now much more elaborated in the revised manuscript (Fig. 2B).

4) Line 125/Fig 2 - "The MPC system is required for survival in macrophages". The title/conclusion of this subsection is not consistent with the data presented. The mutants have a very small effect on survival in macrophages, especially at 24 and 48 hours, where it seems negligible. Confusingly, deletion of the full operon has the least effect. I would urge the authors to reconsider their interpretation of the data.

In this model, the infection occurs in two phases, the bacteria survive for the first 5 hours inside host cells, and after they grow, which can be detected at 24 h and 48 h post-infection (PI) by CFU counting. Our data clearly show >99% mortality ($> 2 \log_{10}$ [CFU] decrease) at 2 h PI, i.e. at the survival phase. Our data suggest that the few mutants that survived to this first stage of the infection are perfectly able to grow inside host cells, as pointed out by the reviewer. This is why we did not conclude about the growth of the various strains. The better survival of the Δmpc operon strain may be due to the fact that, in this strain only, there are no proteins lacking their partners in the Mpc system. We hypothesize that an incomplete system could induce toxic effects and explain their lower survival compared to the Δmpc operon strain. Alternatively, we cannot exclude that the Δmpc operon strain already acquired suppressive mutations decreasing the penetrance of the phenotype.

5) Previous work has suggested a difference in OMV production in Mce mutants (PMID: 26806181). Since the authors are characterizing OMVs, was this also observed? How does any change in the quantity of OMVs impact the lipidomics analysis and what would the authors expect the effect to be if there is a difference in total OMV production?

We indeed observed a slight increase in OMV abundance in the Δmpc mutants compared to the wild type strain (see answer to comment 2 of reviewer #2). This effect is much more subtle than the one described previously by Grasekamp *et al.* We analyzed our data in both absolute and relative values, and our observations are consistent with both analyses. In the revised manuscript, we provide the absolute values to answer to comment 3 of reviewer #2.

6) I did not follow the logic of the following conclusion, it is possible that I missed something, and would appreciate clarification (lines 137-145): because mce operon mutants have less cardiolipin and similar levels of PLs in OMVs, this system may act as a PL transport system

As you will see in the revised manuscript, we re-interpreted our conclusion regarding this analysis. However, the lower abundance of CL in OMV remains, indeed. This result can be interpreted in two ways, either Mpc is a transporter with a preference for CL and thus this absence of the Mpc system induces a lower concentration of CL in the OM and in the OMV, or Mpc is transporting PL (more than CL) back in the IM from the OM, and thus its absence generates an excess of PL in the OM and the OMV, and thus a lower proportion of CL in the OM and possibly the OMV. We think that our data cannot help to determine the directionality of the transport of a specific nature of lipids, this is why we avoid discussing this in our revised manuscript.

7) Use of AlphaFold models, and conclusions drawn. while AF predictions have revolutionized our ability to understand biology, they are hypothesis generators, not pieces of data. As such, it is imperative that conclusions based on AF predictions are validated experimentally. As evidenced by Fig. S7, S8 and the discussion from the authors, many different AF predictions are generated for this particular complex. If the authors would like to make hard conclusions from these (such as bridging of the IM and OM, oleic-acid mediated conformations, hydrophobicity, etc.) then further experimental data are needed to validate predictions, test hypotheses, and arrive at conclusions. If, on the other hand, the authors choose not to perform followup experiments and validation, then the interpretation and conclusions need to be tempered, and largely moved to the discussion, as they are valuable but speculative. As an example, Line 168 "Mpc bridges inner to outer membrane" - there are no data speaking to this conclusion. The only thing presented here is an alphafold model - it would be reasonable to speculate that this is a possibility in the discussion. However, even an experimental structure outside the context of inner and outer membranes would not be sufficient to draw a conclusion with that level of certainty.

As indicated in the previous comments, we corrected several interpretations that were indeed overstatements in the original manuscript. Our structural predictions are not solely based on Alphafold models, but also on the homology between Mpc proteins and components of the Mla and Pqi systems, now more clearly reported in Appendix Figure S5-S6. The bridge between both membranes is based on the localization of the MpcA to the outer membrane (OM) (which is very likely but not proven, indeed) and the association of MpcD and MpcE to the inner membrane, which is attested by transmembrane segments and homology to the Mla system.

8) Fig 5 - some inconsistency between the legend and the figure. It is not clear what the break in the AlphaFold predictions are towards the lipoprotein. The coloring on the Mce protein of B is not explained, and the key is not explained, and the interior shown does not look hydrophobic, if the colors are supposed to be a hydrophobicity scale. Since this is an AlphaFold prediction, discussing hydrophobicity (which will depend on rotamer conformation) may or may not be accurate.

We apologize if the figures were not clear enough in the original manuscript. The Fig. 5 of the original manuscript is modified in the revised manuscript (now Fig. 6), taking into account a suggestion of reviewer #2 regarding the modeling of the complex. In Fig. 6B, the hydrophobicity is shown for MpcD only, the red color indicating the predicted hydrophobicity, while white/grey depicts hydrophilic regions. The interior of the channel is almost completely red because side chains of residues pointing inside the channel are almost all hydrophobic (mainly Leu, Ile and Phe).

9) The *V. Parvula* system shown in Fig. 6 is much more similar to the current system than the Mla system, and both are TGD-like systems - it is unfortunate that the authors named it Mla, but for the readers of the paper, and the field, this is likely to be confusing. It also lacks an MlaC homolog, as mpcEFDA does.

We guess that the *V. parvula* system was named according to the function that they proposed, i.e. the transport of PLs back to the IM. As pointed out above, the different systems diverge by their anchoring to the OM: the C-terminus of Mce1, a beta-barrel in the OM, and here and in the Pqi system, an octameric lipoprotein. We thus name our system according to the preexisting literature, and the homologies to Mla and Pqi systems, which is supported by Appendix Figure S5-S6 in the revised manuscript.

Dear Xavier,

Thank you again for submitting your revised manuscript (EMBOJ-2024-119476R) to The EMBO Journal for our consideration, and for your patience during peer review. As I have already informed you, your revised manuscript has now been seen by the three original referees who had previously assessed the first version of your manuscript, and we have received their comments, which are included below. I am very pleased to say that the referees find the revised manuscript significantly improved and strengthened, addressing the majority of the initially raised criticisms and concerns. As you will see, referees #1 and #2 have no further comments and recommend publication of the manuscript, while referee #3 has two remaining points that we would kindly ask you to textually address in a final version of your manuscript. Please include in your resubmission a brief point-by-point response explaining how these two concerns are addressed in the revised manuscript.

From the editorial side, there are also a few changes and corrections we need you to make in the final version of your manuscript, before we can accept it for publication in The EMBO Journal:

- Please deposit to appropriate repositories all DNA sequencing and mass spectrometry data produced in your study, and provide in the Data availability statement of your revised manuscript all relevant information: the database, ID and permanent specific URL of each dataset. Please make sure that all data will be publicly available at the time of publication. The reviewer access information can now be removed from the Data availability section.
- The corresponding author needs to be indicated and his e-mail address must be provided on the title page of the manuscript.
- The funding information included in the Comments box could not be extracted by our production team, and therefore all funders should be added to the "More Funders" list, please.
- Please note that no more than 5 keywords can be listed (you currently have 7).
- Please include the heading "References" before the list of citations in the revised manuscript file.
- Please include a conflict-of-interest statement in your revised manuscript, with the heading "Disclosure and competing interests statement", according to our instructions here:
<https://www.embopress.org/page/journal/14602075/authorguide#conflictsofinterest>.
- The author contributions statement should be removed from the manuscript file. Instead, we use CRediT to specify the contributions of each author in the journal submission system. Please feel free to use the free text box to provide more detailed descriptions during submission. See also our guide to authors for more information:
<https://www.embopress.org/page/journal/14602075/authorguide#authorshippinguidelines>.
- As per our journal's policy, "data not shown" (on pages 19, 20, and 21) is not permitted. All data referred to in the paper should be displayed in the main or Expanded View figures, or in the Appendix. Please add these data or change the text accordingly if these data are not central to the study and its conclusions, or properly cite the respective published sources if these data can be found elsewhere.
- All Figure panel callouts should be listed sequentially.
- We noticed that callouts for missing callouts for Fig. 4B are missing.
- The header of the title page of your Appendix file should be "Appendix for" followed by the manuscript title. Please also include on the same page a Table of Contents including the page numbers for the listed items.
- The legends of the Appendix Tables and Figures should only be provided in the Appendix file - please remove them from the main manuscript file.
- Thank you for providing the requested Source Data (SD). Please note that the SD for Fig. 2 should be in a zip folder, and Figure panels should be clearly labeled. If all requested SD for Fig. 8 are included in the deposited dataset in the PRIDE database, please add this clarification in the comments box at the bottom of your Source Data checklist.
- Please note that EMBO press papers are accompanied online by:
 - A) a short (2 sentences) summary of the findings and their significance,
 - B) 2-5 short bullet points highlighting the key results, and
 - C) a synopsis image in .jpg or .png format that is exactly 550 pixels wide and 300-600 pixels high (the height is variable). Please note that the text needs to be legible at the final size.Please upload this information along with your revised manuscript (the text for A and B should be provided in a separate Word

file).

- During our standard Figure checks, we detected:

1. possible re-use of blots in Figure 5B, and
2. possible re-use of blots between Figure 5B and Figure EV2A.

We kindly request you to check these Figures carefully, correct them if necessary (in which case please include an explanation in your cover letter), or explain if this reuse is intentional and experimentally justified (in which case, please clearly mention the reuse in the respective Figure legends of the revised manuscript). Please also make sure to include the uncropped, original blots for these Figures in your Source Data.

- During our routine pre-acceptance checks, our data editors have raised the following queries regarding Figures, data, and legends. Please make sure that the following requests are fully addressed in the final version of your manuscript:

1. Please note that the exact p values are not provided in the legends of Figures 3A, 4B, 5C.
2. Please indicate the statistical test used for data analysis in the legend of Figure 4B.
3. Please note that in Figure 3A there is a mismatch between the annotated p values in the Figure legend and the annotated p values in the Figure file that should be corrected.
4. Please note that information related to "n" is missing in the legend of Figure EV1 B.
5. Please note that the error bars are not defined in the legend of Figure EV1 B.

- The order of the manuscript sections must be corrected as follows: Title page - Abstract and Keywords - Introduction - Results - Discussion - Methods - Data Availability - Acknowledgements - Disclosure and Competing Interests Statement - References - Figure Legends - main Tables (if there are any) - Expanded View Figure Legends.

Please also note that as part of the EMBO publications' Transparent Editorial Process, The EMBO Journal publishes online a Peer Review File along with each accepted manuscript. This File will be published in conjunction with your paper and will include the referee reports, your point-by-point response and all pertinent correspondence relating to the manuscript. You can opt out of this by letting the editorial office know (contact@embojournal.org). If you do opt out, the Peer Review File link will point to the following statement: "No Peer Review File is available with this article, as the authors have chosen not to make the review process public in this case."

We look forward to seeing a final version of your manuscript as soon as possible. Please let us know if you have any questions and use this link to submit your revision: <https://emboj.msubmit.net/cgi-bin/main.plex>.

Best regards,

Ioannis

Referee #1:

I am satisfied with the revision. Although not all of the newly performed experiments were successful, the authors tried their best to respond to the critical reviewer comments.

Referee #2:

The authors made important efforts to answer the concerns of the reviewers. I have the feeling that the manuscript is now suitable for publication.

Referee #3:

We thank the authors for their responses and the revised manuscript. Some comments have been addressed, and we agree with the decision to remove data that was not rigorous and reproducible.

1) Regarding the similarities to Mce1 and TGD2, perhaps the original comment was not clear enough. It was not just a matter of mentioning these systems in the introduction, which had indeed previously been overlooked. The point is that the overall structure of the MCE proteins in these systems, coupled with their function together with ABC transporters makes them very similar to the system being studied in this manuscript. At the outer membrane, many of the homologous systems show variations, and it is currently unclear how any of them (except Mla) function at the outer membrane, at the molecular/mechanistic level. However, the parallels between the overall MCE protein architecture and function together with ABC transporters are very clear. In this aspect, Mce1 (mycobacteria), TGD2 (arabidopsis) and Mpc (current study) are all variations on the same theme. As such, the other systems form a very relevant basis for comparison. The authors' conclusions regarding the homology of the Mpc system to Mla but not TGD/Mce1 is also quite confusing, since the part of the Mpc system that is homologous to Mla is MpcEF, the ABC transporter, which is also homologous to the TGD systems and Mce1.

3) Regarding the title of the manuscript, and corresponding experiments, in the response to reviewers, the authors note speculative aspects of their results, including "We hypothesize that an incomplete system could induce toxic effects and explain their lower survival compared to the Δ mpc operon strain. Alternatively, we cannot exclude that the Δ mpc operon strain already acquired suppressive mutations decreasing the penetrance of the phenotype." The two phases of infection are also a model, as noted. The speculations are interesting, but what is a bit unsettling is that it is hard to tell apart what is conclusive based on experimental results, and what is speculation/may have alternate interpretations. Conclusions should be restricted to what is unambiguously shown by the data, caveats to the conclusions or alternate interpretations should be clearly noted, and speculation, while certainly valuable, should be in the discussion section.

Response to reviewer 3

Manuscript EMBOJ-2024-119476R

1) Regarding the similarities to Mce1 and TGD2, perhaps the original comment was not clear enough. It was not just a matter of mentioning these systems in the introduction, which had indeed previously been overlooked. The point is that the overall structure of the MCE proteins in these systems, coupled with their function together with ABC transporters makes them very similar to the system being studied in this manuscript. At the outer membrane, many of the homologous systems show variations, and it is currently unclear how any of them (except Mla) function at the outer membrane, at the molecular/mechanistic level. However, the parallels between the overall MCE protein architecture and function together with ABC transporters are very clear. In this aspect, Mce1 (mycobacteria), TGD2 (arabidopsis) and Mpc (current study) are all variations on the same theme. As such, the other systems form a very relevant basis for comparison. The authors' conclusions regarding the homology of the Mpc system to Mla but not TGD/Mce1 is also quite confusing, since the part of the Mpc system that is homologous to Mla is MpcEF, the ABC transporter, which is also homologous to the TGD systems and Mce1.

We totally agree with this comment and this suggestion. The comparison with the sequences of the systems in *Mycobacterium* and *Arabidopsis* is now indicated in the Results, and the sequence alignments are now provided in Appendix Fig. S5. The common feature of the Mla, TGD and Mpc systems, the ABC transporter, is now cited in the Discussion, as well as the conserved needle structure found in TGD complexes.

3) Regarding the title of the manuscript, and corresponding experiments, in the response to reviewers, the authors note speculative aspects of their results, including "We hypothesize that an incomplete system could induce toxic effects and explain their lower survival compared to the Δmpc operon strain. Alternatively, we cannot exclude that the Δmpc operon strain already acquired suppressive mutations decreasing the penetrance of the phenotype." The two phases of infection are also a model, as noted. The speculations are interesting, but what is a bit unsettling is that it is hard to tell apart what is conclusive based on experimental results, and what is speculation/may have alternate interpretations. Conclusions should be restricted to what is unambiguously shown by the data, caveats to the conclusions or alternate interpretations should be clearly noted, and speculation, while certainly valuable, should be in the discussion section.

In the previous response to the reviewers, when we wrote "In this model", it was meaning "In this infection model", since the experiments were performed with cell lines and not the natural host. The two phases of infection are an observation made independently by several laboratories (including the well-recognized teams of J.-P. Gorvel, J. Celli and S. Salcedo), and it is now indicated in the revised manuscript, together with two independent bibliographic references. Compared to standards in the field, our data convincingly demonstrate that the *mpc* operon is required for survival at short times post-infection. We generated 4 deletion strains, all complemented, that could have additional problems like the toxicity of an incomplete system, but this does not rule out the requirement of the Mpc system for survival at short times post-infection. In the new manuscript, we added a sentence in the discussion to propose an additional interpretation of the data of Fig. 3A, as indicated in the response to the reviewer. We also replaced "essential" by "required" in the title of the manuscript.

Dear Xavier,

Congratulations on an excellent study! I am very pleased to inform you that your manuscript has been accepted for publication in The EMBO Journal. Thank you for comprehensively addressing the initially raised referees' criticisms and concerns, as well as all editorial requests for changes and corrections.

If you have any questions, please do not hesitate to contact the Editorial Office. Thank you for your contribution to The EMBO Journal. Working with you has been a pleasure!

Best regards,

Ioannis
